# RAN translation at *C9orf72*-associated repeat expansions is selectively enhanced by the integrated stress response

Katelyn M. Green[1,2], M. Rebecca Glineburg[1], Michael G. Kearse[1,3], Brittany N. Flores[1,2], Alexander E. Linsalata[1,2], Stephen J. Fedak[1], Aaron C. Goldstrohm[4], Sami J. Barmada[1] & Peter K. Todd[1,5]

Repeat-associated non-AUG (RAN) translation allows for unconventional initiation at disease-causing repeat expansions. As RAN translation contributes to pathogenesis in multiple neurodegenerative disorders, determining its mechanistic underpinnings may inform therapeutic development. Here we analyze RAN translation at $G_4C_2$ repeat expansions that cause *C9orf72*-associated amyotrophic lateral sclerosis and frontotemporal dementia (C9RAN) and at CGG repeats that cause fragile X-associated tremor/ataxia syndrome. We find that C9RAN translation initiates through a cap- and eIF4A-dependent mechanism that utilizes a CUG start codon. C9RAN and CGG RAN are both selectively enhanced by integrated stress response (ISR) activation. ISR-enhanced RAN translation requires an eIF2α phosphorylation-dependent alteration in start codon fidelity. In parallel, both CGG and $G_4C_2$ repeats trigger phosphorylated-eIF2α-dependent stress granule formation and global translational suppression. These findings support a model whereby repeat expansions elicit cellular stress conditions that favor RAN translation of toxic proteins, creating a potential feedforward loop that contributes to neurodegeneration.

[1] Department of Neurology, University of Michigan, Ann Arbor, MI 48109-2200, USA. [2] Cellular and Molecular Biology Graduate Program, University of Michigan, Ann Arbor, MI 48109, USA. [3] Department of Biochemistry and Biophysics, University of Pennsylvania, Philadelphia, PA 19104, USA. [4] Department of Biochemistry, Molecular Biology, and Biophysics, University of Minnesota, Minneapolis, MN 55455, USA. [5] VA Ann Arbor Healthcare System, Ann Arbor, MI 48109, USA. M. Rebecca Glineburg, and Michael G. Kearse contributed equally to this work. Correspondence and requests for materials should be addressed to P.K.T. (email: petertod@umich.edu)

Nucleotide repeat expansions cause multiple neurodegenerative disorders[1]. Recently, an unconventional form of translation initiation known as repeat-associated non-AUG (RAN) translation has emerged as a novel mechanism by which repeat expansions cause toxicity[2, 3]. RAN translation occurs in the absence of an AUG start codon, in multiple reading frames, through an expanded repeat to produce homopolymeric or dipeptide-repeat-containing proteins (DPRs). This non-canonical initiation event occurs in multiple disorders, including at CAG and CUG repeats in spinocerebellar ataxia type 8 (SCA8) and Huntington's disease, and at CGG and CCG repeats in fragile X-associated tremor/ataxia syndrome (FXTAS)[2, 4–6].

A $G_4C_2$ repeat expansion located in the first intron of C9orf72 is the most common known inherited cause of both amyotrophic lateral sclerosis (ALS) and frontotemporal dementia (FTD)[7, 8]. In C9ALS/FTD, this repeat is often expanded from <25 units to upwards of several hundred, although disease occurs with as few as 70 repeats[7–9]. Despite its intronic localization, RAN translation occurs at this locus (C9RAN) at both sense strand-derived $G_4C_2$ repeats and antisense strand-derived $C_4G_2$ repeat transcripts to generate six different DPRs[10–12]. These DPRs accumulate in p62 and ubiquitin positive aggregates in C9ALS/FTD neurons, which is consistent with pathology observed in many repeat expansion disorders[10, 11].

DPRs are both necessary and sufficient to induce neurodegeneration in simple model systems[13–15]. DPRs elicit toxicity through a number of mechanisms, including altered ribosomal biogenesis, impaired nucleocytoplasmic transport, shifts in RNA metabolism, protein sequestration, and impaired protein quality control pathways[13, 16–21]. The charged DPRs, glycine–arginine and proline–arginine, in particular accumulate in membrane-less organelles, including RNA granules, and are associated with suppressed global protein synthesis and altered granule dynamics[17, 22–24]. However, most of these findings originated from studies that relied upon DPR production not through RAN translation, but through AUG-initiated translation of a synthetic non-repetitive RNA sequence. As such, while the relative toxicity of different DPR species in isolation is established, their relative stoichiometry and translation kinetics remain unclear.

Despite a potentially central role in multiple neurodegenerative disorders, our understanding of the mechanism(s) of RAN translation is incomplete. Canonical eukaryotic translation initiation follows a scanning mechanism, where the 5′ m7G-cap recruits the cap-binding complex eIF4F (composed of the cap-binding protein eIF4E, eIF4G, and the DEAD box helicase eIF4A) to the 5′ end of the mRNA[25, 26]. In parallel, the multi-subunit GTPase initiation factor eIF2 binds to the initiator methionine tRNA ($tRNA_i^{Met}$) in its GTP-bound state to generate the ternary complex, which then assembles with the 40S ribosomal subunit and other initiation factors to form the 43S pre-initiation complex (PIC). The PIC associates with eIF4F at the mRNA 5′ m7G-cap. This complex scans along the mRNA in a 5′–3′ direction in a process promoted by eIF4A, until it encounters an AUG start codon in an appropriate Kozak sequence context in the P-site[25, 26]. eIF5 then promotes hydrolysis of GTP to GDP on eIF2, and eIF2-GDP and Pi is released, allowing for recruitment of the 60S subunit and decoding of the second codon in the A-site[25–27].

While the scanning model of translation initiation applies to many transcripts under basal conditions, a variety of alternative initiation mechanisms exist that bypass these requirements. Internal ribosomal entry sites (IRESs) are often complex RNA structures that promote translation initiation independent of the 5′ cap, specific initiation factors and, in certain cases (e.g. the cricket paralysis virus [CrPV] IRES), bypass the need for any initiation factors or an AUG codon[26, 28]. In addition, cells actively regulate translation initiation after exposure to a variety of perturbations in cellular homeostasis through the integrated stress response (ISR, reviewed in refs [25, 29, 30]). ER stress, viral infection, amino acid starvation and other triggers stimulate ISR kinase cascades that converge to phosphorylate the regulatory initiation factor eIF2α at serine 51. This phosphorylation event suppresses global protein synthesis by inhibiting eIF2B, the GEF that exchanges GDP for GTP on eIF2, thus preventing eIF2 rebinding to $tRNA_i^{Met}$ and forming additional ternary complexes. However, a subset of mRNAs escapes this suppression through use of upstream open reading frames (uORFs), IRES elements, and/or non-AUG initiation codons and retain expression under stress conditions[25, 30–35].

How canonical and non-canonical translation initiation rules intersect with RAN translational requirements is not yet known. If RAN translation contributes meaningfully to pathogenicity in repeat expansion disorders, then identification of specific factors that selectively favor RAN translation may reveal novel targets for therapeutic development across a range of neurological disorders. Moreover, by identifying what cellular conditions influence RAN translation, we can gain insights into critical disease mechanisms underlying C9ALS/FTD and other neurodegenerative diseases. To these ends, we established a series of C9RAN translation-specific reporters and investigated the mechanisms mediating RAN translation at $G_4C_2$ repeat expansions using both in vitro and cell-based assays. C9RAN translation utilizes a cap-, eIF4E-, and eIF4A-dependent scanning mechanism to initiate translation predominantly at a CUG codon just upstream of the repeat. RAN translation at both CGG and $G_4C_2$ repeats is selectively enhanced by ISR activation and eIF2α phosphorylation. These same disease-causing repeats independently impair global protein synthesis and activate stress granule formation, creating a potential feed-forward loop that drives a toxic cascade towards neurodegeneration.

## Results

**$G_4C_2$ RAN translation levels differ across reading frames.** To determine how RAN translation occurs at $G_4C_2$ repeats, we designed a series of reporters containing the first C9orf72 intron through the $G_4C_2$ repeat, for use in in vitro and cell-based assays[36] (Fig. 1a). This sequence was inserted upstream of a modified NanoLuciferase (NLuc) reporter with its AUG start codon mutated to GGG. A carboxy-terminal 3xFLAG-tag was included for western blot detection and a precision protease (PSP) cleavage site was introduced between the repeat and reporter sequences to allow for efficient release of NLuc from the DPR.

Consistent with published results[36], mutating NLuc's AUG (AUG-NLuc) to GGG (GGG-NLuc) resulted in a >1000-fold reduction in luciferase activity in rabbit reticulocyte lysate (RRL) in vitro translation assays and loss of the major immunoreactive protein detected by western blot (Fig. 1b, c). When C9orf72 intron 1 containing 70 $G_4C_2$ repeats was inserted upstream of this reporter in the glycine–alanine (GA) reading frame (Fig. 1a), there was an ~300-fold recovery of luciferase signal and the appearance of a higher molecular weight species by western blot (Fig. 1b, c). Consistent with initiation upstream or within the expanded repeat, the observed molecular weight of GA-NLuc fusion protein increased proportionally with repeat length (Fig. 1b). Similar results were seen when the reporters were expressed in HEK293 cells and in a distinct in vitro system generated from HeLa cell lysates[37] (Supplementary Fig. 1a–c).

As C9RAN also occurs in the glycine–proline (GP) and glycine–arginine (GR) reading frames, we generated additional reporters for production of these DPRs by inserting one or two nucleotides, respectively, between the repeat and NLuc sequences

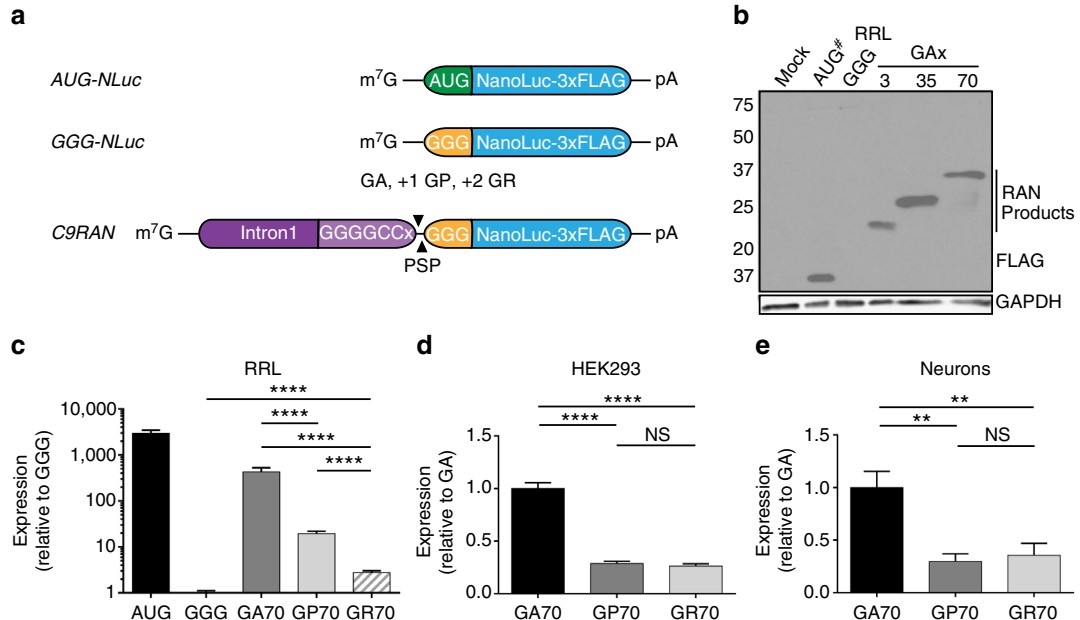

**Fig. 1** C9RAN translation-specific reporters reveal differential expression across reading frames. **a** Schematic of reporters used in this study. C9RAN translation reporters were designed by placing the *C9orf72* intron 1 sequence, including 70 $G_4C_2$ repeats, upstream of a start codon mutant NanoLucifearse (NLuc) and a C-terminal 3xFLAG-tag, in separate reading frames relative to the repeat. **b** Anti-FLAG western blot of control and C9RAN translation reporters expressed in rabbit reticulocyte lysate (RRL). GAPDH is used as a loading control. To prevent over-exposure, the AUG-NLuc control reaction was diluted 1:5 in sample buffer (indicated by [#]). **c–e** Relative expression from C9RAN NLuc reporters (**c**) normalized to GGG-NLuc in RRL ($n = 15$), or normalized to GA-NLuc in (**d**) HEK293 cells ($n = 18$), and (**e**) primary rat hippocampal neurons ($n = 9$). Graph in (**c**) represents mean ± SD. Graphs in **d** and **e** represent mean ± SEM. Two-tailed Student's *t* test with Bonferroni and Welch's correction, **$p < 0.01$; ****$p < 0.0001$. GA glycine–alanine, GP glycine–proline, GR glycine–arginine, PSP precision protease cleavage site

(Fig. 1a). When expressed in RRL, HeLa cell lysate, HEK293 cells, and primary rat hippocampal neurons, C9RAN reporter expression was significantly lower in the GP and GR frames, relative to the GA frame, but still above the GGG-NLuc control (Fig. 1c–e and Supplementary Fig. 1b, c). This difference in NLuc expression between reading frames was likely not a result of differences in protein stability, as the stability of GA, GP, and GR-NLuc fusion proteins were similar in HEK293 cells and not more stable than the AUG-NLuc control (Supplementary Fig. 1d). Additionally, to control for the possibility that each DPR differentially affects NLuc function, we compared luciferase activity of each C9RAN reporter expressed in RRL before and after cleaving at the engineered PSP site (Fig. 1a and Supplementary Fig. 1e, f)[36]. A 15% increase in NLuc activity was observed for the GR-NLuc reporter upon cleavage (Supplementary Fig. 1e), but this small affect cannot account for the nearly 140-fold difference in expression between the GR and GA frames (Fig. 1c). Importantly, this difference in expression level between the three reading frames is consistent with differences in DPR abundance measured in C9ALS/FTD autopsy brain samples by immunohistochemistry[38]. Thus, our C9RAN reporters are specific to each reading frame, exhibit consistent patterns across four systems, and recapitulate the expression pattern seen in disease tissue.

To determine whether differential elongation rates contributed to the observed difference in RAN translation levels across the three sense reading frames, AUG-driven reporters for each reading frame were generated. These reporters contained an AUG start codon in optimal Kozak sequence context immediately upstream of the 70 $G_4C_2$ repeats and lacked the UAG stop codon that natively occurs in the GP reading frame immediately upstream the repeat (Fig. 2a). When expressed in RRL and HEK293 cells, NLuc levels from the GP and GA reporters were no longer significantly different, while GR-NLuc production

remained lower than both (Supplementary Fig. 1g, h). This suggests that the ribosome can synthesize poly-GA and poly-GP products with similar efficiency, but that differences in initiation rates impede poly-GP RAN translation. In contrast, these data indicate that lower synthesis rates of the GR DPR may be caused by differences in both elongation and initiation rates.

**RAN translation at $G_4C_2$ repeats is cap- and eIF4A-dependent.** We next examined the requirement of the 5′ $m^7G$-cap for C9RAN translation by transcribing C9RAN NLuc reporters with either the canonical $m^7G$-cap or an A-cap analog that cannot recruit the cap-binding initiating factor eIF4E, but protects the mRNA from degradation (Fig. 2a). As a control for cap-independent initiation, 5′ $m^7G$- or A-capped mRNAs with the CrPV IRES placed upstream of NLuc were also generated. In RRL and HEK293 cells, A-capped C9RAN reporter mRNAs had dramatically decreased expression in all reading frames compared to $m^7G$-capped mRNAs, whereas translation from the CrPV IRES was unaffected (Fig. 2b and Supplementary Fig. 2a). Similarly, addition of free $m^7G$-cap to the RRL translation reaction in *trans*, to competitively inhibit eIF4E binding to reporter mRNAs, significantly reduced C9RAN in all three readings frames without affecting CrPV expression levels (Fig. 2c). Together, these data indicate that RAN translation from these reporters proceeds through a cap- and eIF4E-dependent mechanism, and that *C9orf72* intron1 with 70 $G_4C_2$ repeats does not act as an IRES.

We next assessed whether C9RAN translation requires ribosomal scanning after recruitment of the PIC to the 5′ $m^7G$-cap. PIC scanning is dependent upon the RNA helicase, eIF4A, which is specifically inhibited by hippuristanol[39]. Addition of hippuristanol to RRL reactions dramatically inhibited translation of the control AUG-NLuc reporter, whereas expression of the CrPV IRES reporter was unaffected (Fig. 2d), which is consistent

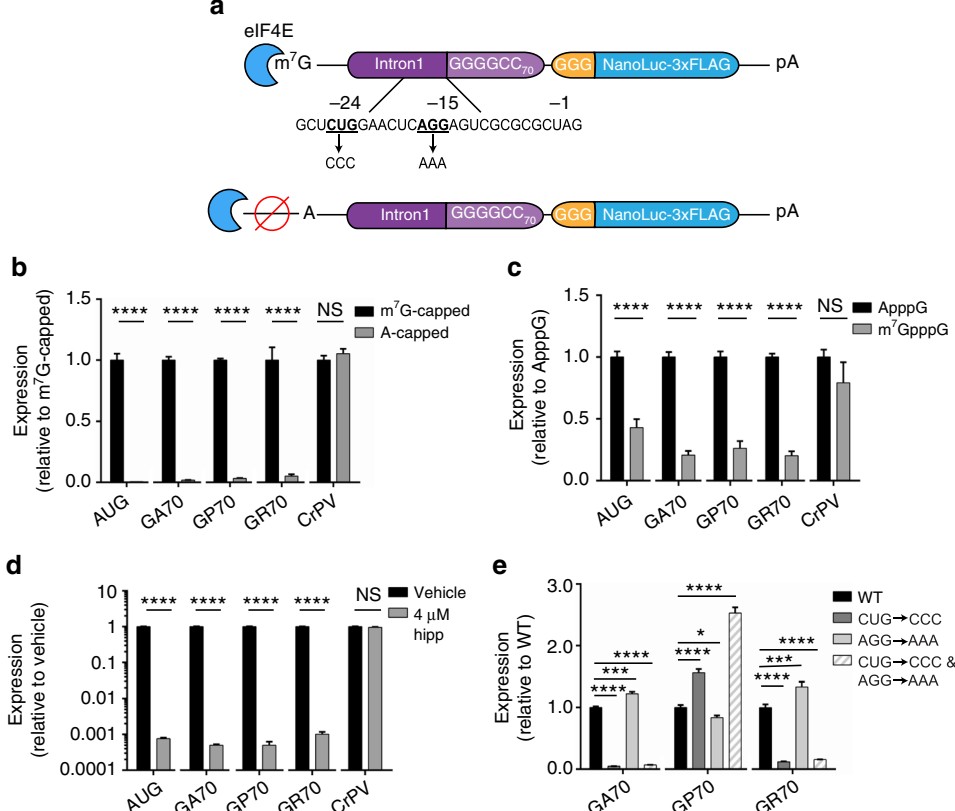

**Fig. 2** C9RAN is cap- and eIF4A-dependent and can initiate at a near-cognate start codon. **a** Schematic of 5′-cap C9RAN reporter mRNAs and near-cognate start codon mutations. **b** Expression of m7G-capped and A-capped control and C9RAN reporters in RRL, $n = 6$. **c** Expression of control and C9RAN reporters in RRL when excess free m7G (250 μM) or equimolar A-cap was added to inhibit eIF4E in *trans*, $n = 6$. **d** Expression of reporter mRNAs when eIF4A, the canonical helicase required for ribosome scanning during initiation, is inhibited with 4 μM hippuristanol (hipp), $n = 6$. **e** Mutational analysis of near-cognate start codons upstream of the repeat in the GA frame as depicted in **a**, $n = 6$. Graphs represent mean ± SEM. Two-tailed Student's *t* test with Bonferroni and Welch's correction, *$p < 0.05$; ***$p < 0.001$; ****$p < 0.0001$. GA glycine–alanine, GP glycine–proline, GR glycine–arginine, CrPV cricket paralysis virus

with previous reports[36, 39]. Expression of all three C9RAN reporters was significantly reduced (over 900-fold) with eIF4A inhibition by hippuristanol (Fig. 2d). Thus, our C9RAN reporters exhibit a strong dependence on eIF4A, similar to what we have previously shown for RAN translation of expanded CGG repeats within the 5′-UTR of *FMR1* in FXTAS[36]. These results are consistent with a scanning model of initiation.

**C9RAN translation uses a near-cognate codon for initiation.** RAN translation at CGG repeats initiates upstream of the repeat at near-cognate codons (codons that differ from AUG by a single nucleotide) in some reading frames[36, 40]. To determine if a similar mechanism occurs in C9RAN, the sequence upstream of the repeat in all three reading frames was examined. In the GA frame, there are two near-cognate codons, as follows: a CUG at position −24 and an AGG at position −15 relative to the first nucleotide of the repeat (Fig. 2a). The CUG codon is in a strong Kozak sequence context, while the AGG codon is not. Mutating the AGG codon to AAA alone had little effect on C9RAN translation in RRL or HEK293 cells for any reading frame (Fig. 2e and Supplementary Fig. 2b). In contrast, mutating the CUG codon to CCC either in the presence or absence of the AGG codon led to a marked reduction in C9RAN in the GA reading frame, in RRL, HEK293 cells, and HeLa cell lysate (Fig. 2e and Supplementary Fig. 2b, c), suggesting that this near-cognate codon is utilized for the majority of RAN translation initiation in the GA frame.

Surprisingly, despite being located in the GA frame, mutating the CUG to CCC also suppressed RAN translation in the GR reading frame in RRL (Fig. 2e), and enhanced RAN translation in the GP reading frame in both RRL and HeLa cell lysate (Fig. 2e and Supplementary Fig. 2c). However, these correlative and anti-correlative effects were not observed in transfected HEK293 cells, where the CUG to CCC mutation did not significantly alter translation in the GR reading frame and decreased translation in the GP reading frame (Supplementary Fig. 2b).

In a reciprocal experiment, we converted this same CUG codon to AUG. This mutation significantly decreased expression in the GP frame in RRL and HEK293 cells (Supplementary Fig. 2d, f). In contrast, the CUG to AUG mutation significantly increased expression in the GR frame in RRL, although this effect was not observed in HEK293 cells (Supplementary Fig. 2d, f). As expected, placing an AUG start codon above the repeat in the GA reading frame greatly enhanced production of GA-NLuc in both systems (Supplementary Fig. 2e, g). Consequently, inhibiting or enhancing translation in the GA frame by modifying start codon usage alters expression in the GP and GR frames. Interestingly, the interplay between translation in the GA and GR frames differs between HEK293 cells and RRL, suggesting functional differences between these assay systems.

Together, these data support a model for C9RAN initiation, in which the PIC is recruited to the mRNA's 5′ cap via interaction with eIF4E and utilizes the eIF4A helicase to scan in the 3′ direction. Initiation at a CUG codon upstream of the repeat is

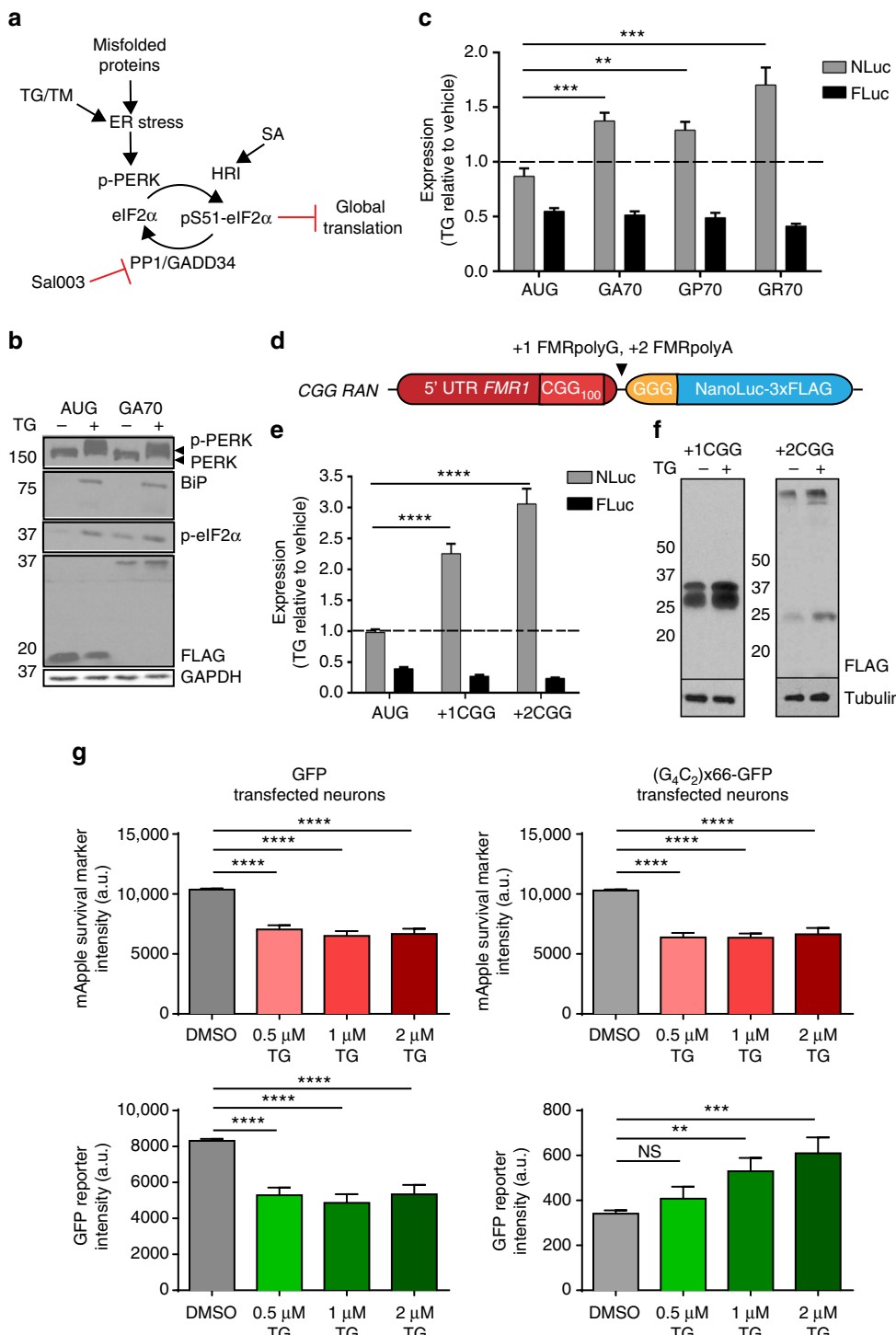

**Fig. 3** RAN translation is selectively activated by the integrated stress response. **a** Schematic of the integrated stress response pathway. **b** Western blot analysis of the ER stress pathway and C9RAN reporter levels in HEK293 cells after treatment with 2 μM TG. GAPDH was used as a loading control. **c** Expression of control and C9RAN NLuc reporters and co-transfected FLuc in HEK293 cells treated with 2 μM TG, $n = 9$. **d** Schematic of the previously published[36] +1 and +2 CGG RAN translation NLuc reporters. **e, f** Expression of control and CGG RAN translation reporters and co-transfected FLuc in HEK293T cells treated with 2 μM TG analyzed by (**e**) luciferase activity, $n = 9$, and (**f**) anti-FLAG western blot. Tubulin was used as a loading control. **g** Fluorescence intensity of mApple and co-transfected GFP (left) or $(G_4C_2) \times 66$-GFP (right) in primary rat cortical neurons, imaged with automated fluorescent microscopy 3 days after treatment with 0.5, 1, or 2 μM TG, $n > 30$. Graphs represent mean ± SEM. Two-tailed Student's $t$ test with Bonferroni and Welch's correction, ** $p < 0.01$; *** $p < 0.001$; **** $p < 0.0001$. PERK endoplasmic reticulum ER-resident kinase, HRI heme-regulated inhibitor kinase, SA sodium arsenite, TG thapsigargin, TM tunicamycin

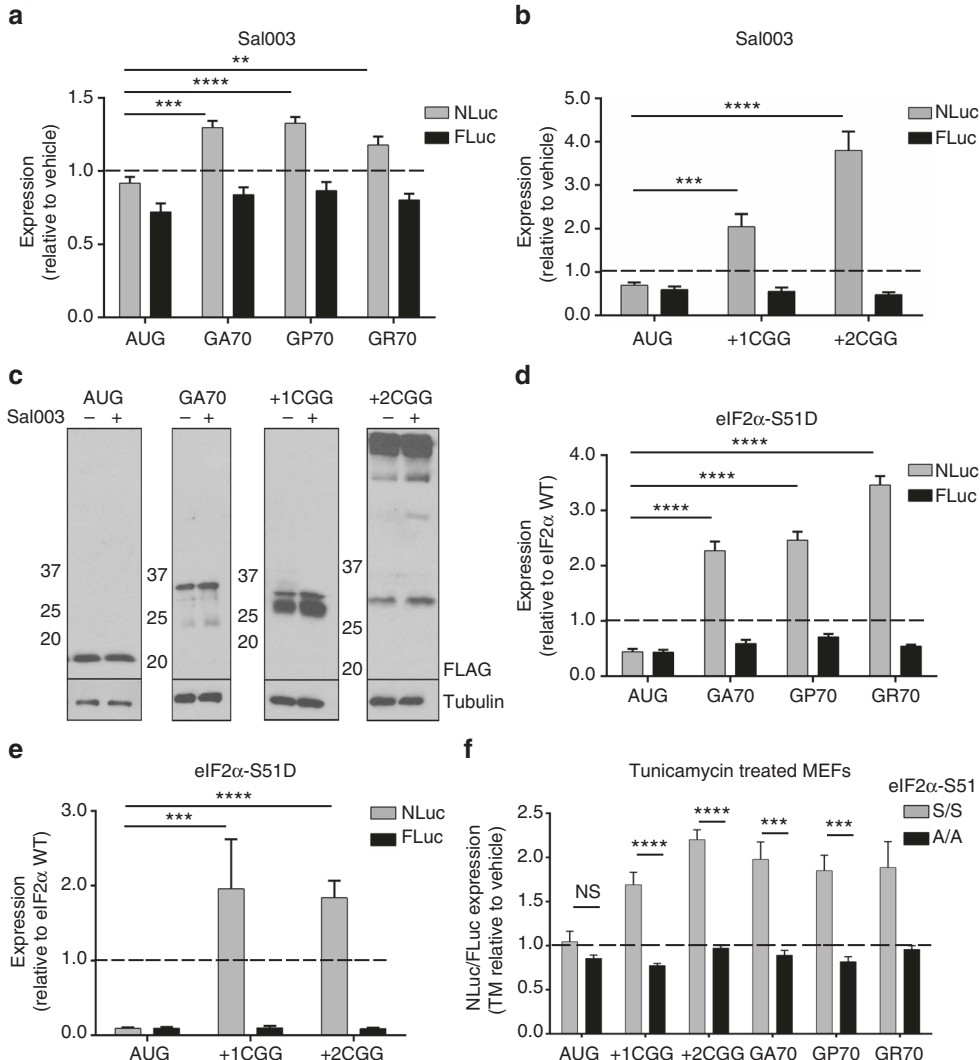

**Fig. 4** RAN translation is resistant to eIF2α phosphorylation. **a** Expression of control and C9RAN NLuc reporters and co-transfected FLuc in HEK293 cells treated with 40 μM Sal003, $n = 6$. **b** Expression of control and CGG RAN NLuc reporters and co-transfected FLuc in HEK293T cells treated with 20 μM Sal003, $n = 9$. **c** Western blot analysis of control, GA70 RAN and CGG RAN NLuc reporters in HEK293T cells treated with 20 μM Sal003. **d** Expression of control and C9RAN NLuc reporters and co-transfected FLuc in HEK293 cells transfected with either WT or S51D (phosphomimetic) eIF2α, $n = 6$. **e** Expression of control and CGG RAN NLuc reporters and co-transfected FLuc in HEK293T cells transfected with either WT or S51D (phosphomimetic) eIF2α, $n = 9-15$. **f** Expression of control, CGG, and C9RAN NLuc reporters normalized to co-transfected FLuc in WT eIF2α-S51 (S/S) and non-phosphorylatable homozygous eIF2α-S51 A/A mutant mouse embryonic fibroblasts (MEFs) following treatment with 1 μg mL$^{-1}$ tunicamycin (TM), $n = 6-9$. Graphs represent mean ± SEM. Two-tailed Student's $t$ test with Bonferroni and Welch's correction, **$p < 0.01$; ***$p < 0.001$; ****$p < 0.0001$

important for translation in the GA reading frame. However, if the ribosome fails to initiate at this CUG codon, it may continue scanning into the repeat, where it could initiate in the GP frame in the absence of any near-cognate codon.

**Cellular stress selectively enhances RAN translation.** Cellular stressors, such as viral infection, misfolded proteins and amino acid starvation, can activate the ISR through one of four kinases (interferon-induced double-stranded RNA-dependent eIF2α kinase [PKR], endoplasmic reticulum [ER]-resident kinase [PERK], general control non-derepressible 2 [GCN2], or heme-regulated inhibitor kinase [HRI]), that all phosphorylate eIF2α at serine 51 (Fig. 3a)[29, 33]. As both start codon stringency and initiation kinetics are modulated in response to eIF2α phosphorylation following ISR activation[30, 31, 33, 41], we hypothesized that RAN translation might be refractory to ISR-activation. To test this, cells transfected with C9RAN reporters were exposed to the ER calcium pump inhibitor, thapsigargin (TG), to cause ER

stress and activate the ISR through PERK (Fig. 3a). As expected, ER stress induction by TG led to PERK phosphorylation, BiP upregulation, and increased eIF2α phosphorylation (Fig. 3b), as well as global translation repression (Supplementary Fig. 3a)[42]. Consistent with this, expression of both the AUG-NLuc and co-transfected firefly luciferase (FLuc), which serves as an independent internal control, decreased when cells were stressed with TG (Fig. 3c). This effect was less pronounced for AUG-NLuc than FLuc, as expected due to its heightened stability (Supplementary Fig. 1d). Therefore, when destabilized with a PEST tag, AUG-NLuc expression was more greatly decreased by TG treatment (Supplementary Fig. 3b). In contrast, expression of all C9RAN translation reporters was significantly increased during ER stress with TG treatment as shown by both luciferase activity and western blot (Fig. 3b, c).

To determine if this enhancement was unique to C9RAN, we also interrogated the effect of ISR induction on CGG RAN translation. RAN translation at CGG repeats occurs

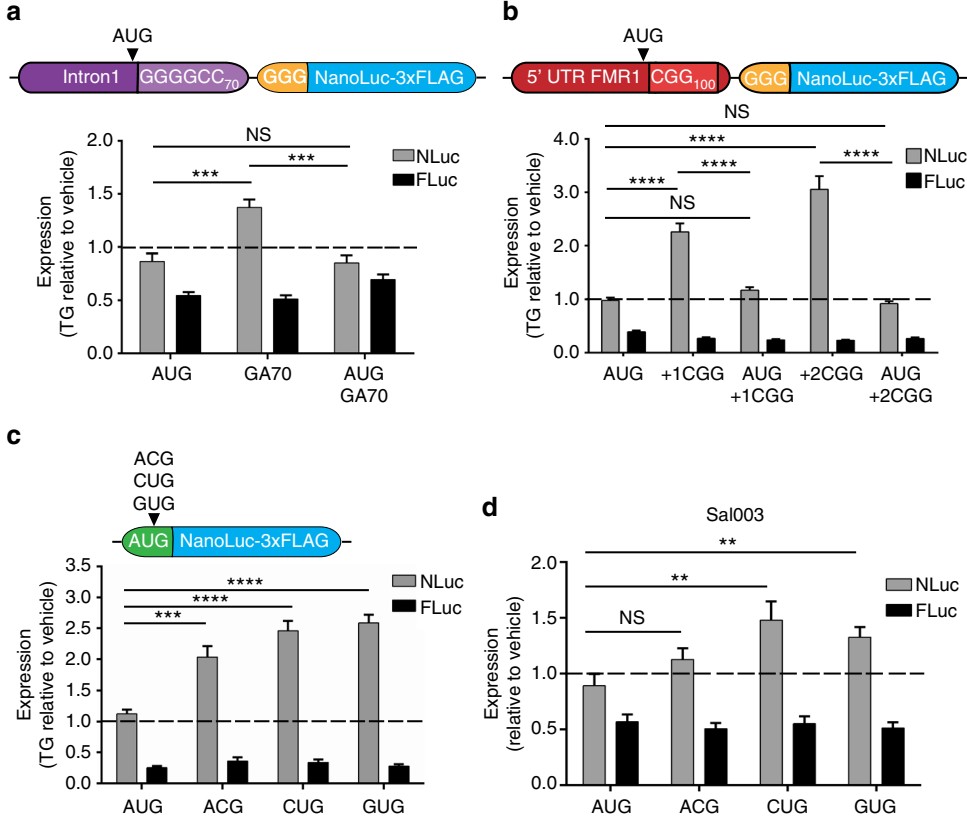

**Fig. 5** Near-cognate codons are sufficient to allow for stress-induced translation. **a** Top: schematic of C9RAN reporter with AUG codon inserted upstream of repeat. Bottom: expression of control, C9RAN NLuc, and AUG-driven C9 reporters and co-transfected FLuc in HEK293 cells treated with 2 µM TG, $n = 6$–9. **b** Top: schematic of CGG RAN reporter with a near-cognate codon mutated to AUG. Bottom: expression of control, CGG RAN NLuc, and AUG-driven CGG reporters and co-transfected FLuc in HEK293T cells treated with 2 µM TG, $n = 9$. **c** Top: schematic showing location of near-cognate codon substitutions made in AUG-NLuc. Bottom: expression of control and near-cognate codon NLuc reporters and co-transfected FLuc in HEK293T cells treated with 2 µM TG, $n = 9$. **d** Expression of control and near-cognate codon NLuc reporters and co-transfected FLuc in HEK293T cells treated with 20 µM Sal003 for 5 h, $n = 12$. Graphs represent mean ± SEM. Two-tailed Student's $t$ test with Bonferroni and Welch's correction, \*\*$p < 0.01$; \*\*\*$p < 0.001$; \*\*\*\*$p < 0.0001$

predominantly in two reading frames (Fig. 3d)[5]. Initiation in the +1 (GGC, glycine) reading frame occurs mainly at either an ACG or GUG codon just upstream of the repeat to generate FMRpolyG, a protein that accumulates in intranuclear inclusions in FXTAS[5, 36, 40]. In contrast, RAN translation in the +2 (GCG, alanine) reading frame is less robust and likely initiates in the repeat sequence itself to produce FMRpolyA[5, 36]. TG-induced ER stress significantly enhanced luciferase activity and immunodetection of both +1 or +2 CGG RAN translation reporters, but not the internal FLuc control, compared to vehicle treatment in HEK293T cells (Fig. 3d–f). Similarly, stress-stimulated global translation attenuation with tunicamycin (TM), which blocks N-linked glycosylation in the Golgi to cause ER stress, or sodium arsenite (SA), which causes oxidative stress and activation of the HRI kinase (Fig. 3a and Supplementary Fig. 3a), also either spared or enhanced CGG RAN translation while significantly inhibiting AUG-initiated translation (Supplementary Fig. 3c, d). Thus, activation of ISR pathways enhances RAN translation across at least two repeats and five separate reading frames.

To assess the effect of ISR activation of C9RAN in neurons, we utilized automated fluorescent microscopy of primary rat cortical neurons co-transfected with GFP or $(G_4C_2)\times66$-GFP (a reporter containing 66 $G_4C_2$ repeats in the GP reading frame just upstream of GFP) and mApple, a red fluorescent marker used to facilitate longitudinal tracking. Single-cell fluorescence intensity for both reporters was measured for 3 days after TG treatment at varying doses. As observed with AUG-initiated luciferase reporters in

HEK cells, ER stress induction reduced signal from AUG-initiated mApple and AUG-initiated GFP (Fig. 3g). However, expression of $(G_4C_2)\times66$-GFP reporter increased in a dose-dependent manner with TG (Fig. 3g). Consequently, cellular stress induction in multiple cell types, including neurons, selectively enhances the production of neurotoxic RAN proteins involved in two distinct neurodegenerative diseases.

**eIF2α phosphorylation selectively enhances RAN translation.** We next explored additional approaches to more directly evaluate the role of eIF2α phosphorylation in RAN translation. Salubrinol (Sal003) selectively inhibits PP1, the major phosphatase that acts on eIF2α (Fig. 3a); treatment with Sal003 thus increases cellular levels of phosphorylated eIF2α[43] (Supplementary Fig. 4a). Addition of Sal003 to transfected cells had only modest inhibitory effects on production of both canonically-translated AUG-NLuc and FLuc reporters (Fig. 4a, b). In contrast, treatment with Sal003 significantly enhanced RAN translation from both C9RAN and CGG RAN reporters, by both luciferase activity and western blot (Fig. 4a–c). Furthermore, co-transfecting cells with NLuc reporters and a phosphomimetic form of eIF2α (S51D) suppressed translation of both AUG-NLuc and control FLuc reporters, relative to co-transfection with WT eIF2α, while selectively enhancing RAN translation from C9 and CGG RAN reporters in all assessed reading frames (Fig. 4d, e). These data show that eIF2α phosphorylation is sufficient to enhance RAN translation.

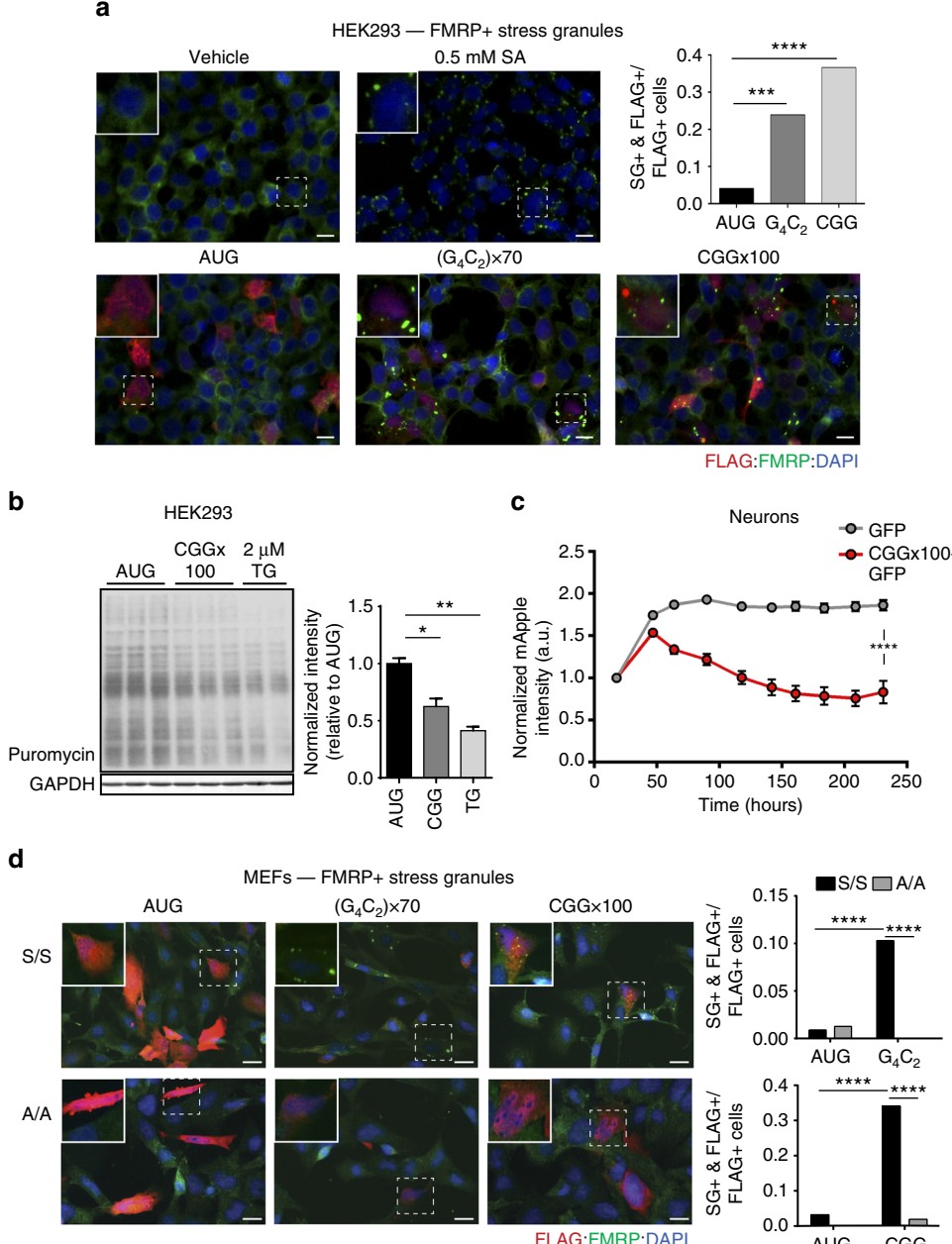

**Fig. 6** CGG and $G_4C_2$ repeat expansions induce phosphorylated-eIF2α-dependent stress granules. **a** Top left: immunofluorescent images of HEK239 cells treated with vehicle or 0.5M SA. Bottom: immunofluorescent images of HEK239 cells expressing control, $(G_4C_2)$×70, or CGG×100 reporters, scale bar = 100 μm. Top right: quantification of the proportion of FLAG-positive cells with FMRP-positive stress granules (SGs) for each genotype, $n > 70$. **b** Western blot and quantification of puromycin incorporation in cells transfected with control AUG-NLuc or CGG×100 reporter, or treated with 2 μM TG as a positive control. GAPDH is used as a loading control. Graph represents mean ± SEM. **c** mApple fluorescent intensity in primary rat cortical neurons co-transfected with GFP or CGG×100-GFP, longitudinally imaged with automated fluorescent microscopy for 10 days following transfection, $n > 68$. Graph represents mean ± 95% confidence interval. **d** Left: immunofluorescent images of WT eIF2α- S51 S/S and eIF2α- S51 A/A MEFs expressing control, $(G_4C_2)$×70, or CGG×100 reporters, scale bar = 100 μm. Right: quantification of the proportion of FLAG-positive cells with FMRP-positive SGs for each genotype, $n > 40$. FLAG marks reporter expressing cells, FMRP mark SGs. For **a** and **d**, Fisher's exact test, ***$p < 0.001$; ****$p < 0.0001$. For **b** and **c**, two-tailed Student's $t$ test with Bonferroni and Welch's correction, *$p < 0.05$; **$p < 0.01$; ****$p < 0.0001$

To determine if eIF2α phosphorylation is also necessary for stress-induced RAN translation, MEFs homozygous for the WT (S51 S/S) or a non-phosphorylatable eIF2α (S51 A/A)[42], were co-transfected with CGG or C9RAN NLuc reporters and a FLuc internal control, and treated with TM or TG. While both TM and TG treatments increased CGG and C9RAN NLuc expression relative to FLuc in WT MEFs, this enhancement was lost in the S51 A/A MEFs (Fig. 4f and Supplementary Fig. 4b). Thus, eIF2α phosphorylation following

induction of the ISR is both necessary and sufficient to selectively enhance RAN translation under conditions that simultaneously suppress global canonical translation initiation.

**Stress-induced RAN translation requires a non-AUG codon.** Cellular stress and ISR activation can favor initiation at near-cognate codons[30, 32, 33, 35]. To determine if the initiation codon is

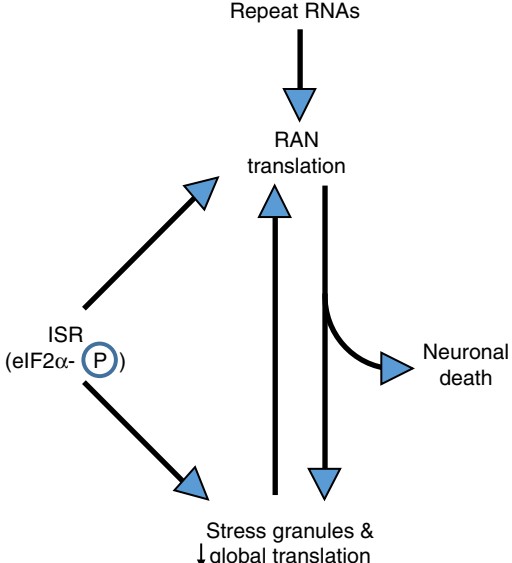

**Fig. 7** Working model for how a feed-forward loop activates RAN translation and cellular stress pathways. Repeat expansions trigger RAN translation. RAN proteins or the repeat RNAs themselves then elicit stress granules and suppress global protein synthesis in a phosphorylated-eIF2α-dependent manner. Activation of the integrated stress response (ISR) and phosphorylation of eIF2α, either by the repeat RNAs or RAN proteins directly or through exogenous cellular stress, can further trigger stress granule formation and suppress global translation while selectively enhancing RAN translation. This creates a feed-forward loop that can contribute to neuronal dysfunction and death

important in ISR-mediated activation of RAN translation, we inserted an AUG start codon upstream of the repeat in C9RAN and CGG RAN reporters[36] to drive canonical translation of the expanded repeat (Fig. 5a, b). Unlike RAN translation, translation of the repeats from a canonical AUG start codon did not show enhancement in response to treatment with TG or Sal003, but behaved similarly to AUG-NLuc (Fig. 5a, b and Supplementary Fig. 5a). Next, to determine if a near-cognate codon alone was sufficient to allow initiation in the setting of cellular stress, we created a set of reporters with the AUG codon of NLuc mutated to one of the near-cognate codons utilized for C9 and CGG RAN translation (Fig. 5c). Mutation to ACG, CUG, or GUG significantly impaired translation under basal conditions compared to AUG (Supplementary Fig. 5b)[44]. However, initiation at near-cognate codons was enhanced in response to treatment with TG and Sal003 (Fig. 5c, d), and was relatively spared when compared to AUG-NLuc when co-transfected with eIF2α S51D (Supplementary Fig. 5c) or treated with SA and TM (Supplementary Fig. 5d, e). Therefore, initiation at a non-AUG start codon is necessary to promote RAN translation in response to stress.

**Repeats trigger stress granules and inhibit global translation**. The ISR is activated in multiple neurodegenerative disorders[45]. C9RAN DPRs can suppress global translation, and overexpression of GA DPR proteins elicits ER stress in neurons[23, 24, 46–48]. We therefore evaluated whether and how repeat-containing constructs impact global protein synthesis and activate the ISR. A common phenomenon during cellular stress is the formation of stress granules, membrane-less structures composed of mRNAs, stalled translation pre-initiation complexes, and multiple RNA-binding proteins (e.g. FMRP and G3BP)[49]. When either $G_4C_2$ or CGG repeat-containing reporters were overexpressed in HEK293

cells, they elicited cytoplasmic FMRP and G3BP-positive stress granule formation (Fig. 6a and Supplementary Fig. 6a)[48]. Concomitantly, overexpression of the +1 CGG RAN reporter inhibited global translation in HEK293 cells, relative to AUG-NLuc, as measured by reduced puromycin incorporation through the surface sensing of translation (SUnSET) assay (Fig. 6b)[50]. These results are consistent with the previously reported translational suppression stimulated by $G_4C_2$ repeats[48].

To determine if similar effects were observed in cells directly impacted in the disease state, we assessed whether CGG repeat overexpression affected translation in primary rat cortical neurons tracked by automated fluorescence microscopy. Neurons were co-transfected with mApple and either a reporter containing 100 CGG repeats upstream of GFP (CGG×100-GFP) or GFP alone. mApple expression was then measured over a 10-day time course of imaging to determine if expanded CGG repeats caused translation attenuation. mApple fluorescence intensity remained stable in neurons co-transfected with GFP alone, but decreased by nearly 50% in neurons expressing CGG×100-GFP (Fig. 6c). This was independent of cytotoxicity elicited by the repeat expansions (data not shown).

Stress granules induced by ISR activation are dependent on eIF2α phosphorylation[42]. To investigate whether $G_4C_2$ and CGG repeat-induced stress granules require phosphorylation of eIF2α, reporters were transfected into eIF2α S51 S/S and S51 A/A MEFs[42]. In agreement with previous studies[42], S/S MEFs formed robust FMRP and G3BP1-positive stress granules in response to TG, while A/A MEFs did not (Supplementary Fig. 6b). Similarly, both $G_4C_2$ and CGG repeat reporters readily induced stress granule formation in S/S MEFs, but exhibited an ~10-fold decrease in stress granule formation in eIF2α A/A MEFs (Fig. 6d and Supplementary Fig. 6c). Together, these data suggest that expression of both $G_4C_2$ and CGG repeat expansions impairs global translation and stimulates the formation of phosphorylated-eIF2α-dependent stress granules.

## Discussion

RAN translation from repeat expansions contributes significantly to the pathology of multiple neurodegenerative disorders, including C9ALS/FTD[2, 5, 13–15, 18, 23, 40, 47]. Here we find that C9RAN translation initiates through a 5′ $m^7G$-cap-, eIF4E-, and eIF4A-dependent mechanism. RAN translation starts either at a near-cognate CUG start codon just upstream of the repeat or potentially within the repeat itself, depending on the reading frame. RAN translation at both $G_4C_2$ and CGG repeats is enhanced by activation of ISR pathways, which normally suppress global translation. This effect is independent of the stress stimuli applied, as it can be recapitulated directly by altering eIF2α phosphorylation, but is dependent upon the initiation codon (i.e. AUG vs. a near-cognate codon). Moreover, overexpression of either CGG or $G_4C_2$ repeats impairs global translation and induces stress granules in an eIF2α phosphorylation-dependent manner. Thus, repeat expansions can trigger a feed-forward loop that drives RAN translation while impairing global translation and altering RNA metabolism (Fig. 7). In the context of the data implicating RAN translation products in the pathogenesis of both FXTAS[5, 40] and C9ALS/FTD[13–15, 20, 23, 47], our findings support a model, whereby an inefficient translation mechanism such as RAN translation might meaningfully contribute to neuronal dysfunction and death in disease.

The $G_4C_2$ repeat containing transcripts studied here are capped and polyadenylated linear mRNAs, but the repeat normally resides within an intron in *C9orf72*. The exact RNA species that undergoes RAN translation in C9ALS/FTD has not been determined empirically. $G_4C_2$ repeat expansions can trigger intron

                                                                9

retention, altered transcription initiation, as well as premature transcription termination, all of which could generate repeat-containing linear mRNAs subject to both 5′ m⁷G-capping and polyadenylation[9, 51, 52]. Moreover, recent data suggest that mRNAs containing the repeat within a retained intron are trafficked to the cytoplasm more efficiently due to interactions with the RNA-binding protein and nuclear export adapter SRSF1[53]. Given the strong cap-dependence we observe with our C9RAN reporters, our data argue that such events, even if rare, could significantly enhance RAN translation efficiency.

The requirements for C9RAN translation initiation closely mirror those previously described for CGG repeats in FXTAS, including initiation at near-cognate codons located 5′-proximal to the repeat in the most robustly-translated reading frames (GA for C9RAN and +1 FMRpolyG reading frame for CGG RAN)[36]. However, here we show that altering initiation levels in one reading frame, by modifying the initiating CUG codon in the GA frame, also alters translation levels in the other two reading frames. In in vitro systems, removing the CUG codon enhances production in the GP reading frame but reduces translation in the GR reading frame. This increase in the GP frame suggests that it competes for initiation with the GA reading frame, and, and, based on the scanning model of translation initiation[25], which the CUG codon is located upstream of the GP initiation site. A UAG stop codon is positioned immediately upstream of the repeat in the GP frame, suggesting GP initiation occurs within the repeat sequence itself. However, alternative possibilities, such as stop codon read-through or frame-shifting upstream of the stop codon remain to be explored. In contrast, the impaired production in the GR reading frame may be consistent with either a +2 or −1 nucleotide frameshift from the GA to the GR reading frame[54]. This is intriguing given evidence linking G-quadruplex structures to −1 ribosomal frameshifting[55]. Such translational frameshifts can occur at other nucleotide repeats[54], although frameshifts are not the dominant cause of RAN translation across different frames at other repeats[2, 5]. Given that each repeat's surrounding sequence context and RNA structure may confer different constraints on RAN initiation, generalizing these findings to RAN initiation at C₄G₂, CCG, CAG and CUG repeats may not be possible without further studies.

Interestingly, the mRNA reporters used in these studies can generate a GA product from only 3 or 35 G₄C₂ repeats, suggesting that the CUG codon, in good Kozak context, does not require an expanded repeat for use by the initiating ribosome. This is consistent with a report finding sparse, neuronal DPR inclusions in a cognitively normal 84 year old woman harboring 30 C9orf72 G₄C₂ repeats[56]. The absence of DPR accumulation in individuals with normal repeat sizes (<25) may indicate that these smaller species are rapidly cleared by cells, or that proper splicing and degradation of the intronic sequence containing the G₄C₂ repeat precludes its translation.

Our results indicate that the non-AUG initiation utilized by RAN translation is critical for its enhancement under stress conditions. We also observe that initiation at near cognate codons in the absence of any repeat sequence can be enhanced by some forms of ISR activation, suggesting that such codons may be sufficient for stress-induced initiation. However, our RAN reporters consistently demonstrate more robust ISR activation than the near-cognate codon reporters lacking a repeat. This suggests that repeats enhance initiation during stress, possibly by creating a blockade for scanning 40S ribosomes that increases near-cognate codon initiation[44, 57]. There may thus be functional overlap between RAN translation and the translational mechanism used by single-stranded alphaviruses, where a hairpin structure within the coding region just 3′ to the start codon maintains active translation in the setting of eIF2α

phosphorylation[58]. Additionally, for at least two stress-enhanced RAN events (GP for G₄C₂ repeats and +2 FMRpolyA for CGG repeats) initiation likely occurs within the repeat in the absence of any near cognate codon[36]. Thus, alternative modes of initiation may depend on the repeat structure to bypass canonical translational control mechanisms and respond to cellular stress pathways.

Recent data from ribosome profiling studies suggest that initiation at near cognate codons may be much more common than previously appreciated[59, 60], and our findings delineate a specific role for near cognate codons in RAN translation. We also observe a role for eIF2 in RAN translation initiation, but the relationship between eIF2α-phosphorylation and start codon fidelity is complicated. ISR activation reduces mature GTP-eIF2-tRNAᵢ^Met ternary complex availability[25]. This can trigger leaky scanning, where uORFs in a poor Kozak sequence context are bypassed for initiation or re-initiation to allow for enhanced translation from the main ORF[30, 31, 34, 61, 62], suggesting that ISR activation enhances start codon fidelity[62]. In contrast, and in agreement with our own findings, translation initiation at near-cognate codons can be enhanced by ISR activation[31, 33, 35]. For example, translation of uORFs that initiate from UUG and CUG start codons in the transcript encoding the ER stress chaperone protein BIP are maintained under stress conditions by utilizing non-canonical initiation factors[33]. Several initiation factors have been implicated in ISR-resistant translation, including eIF2D, eIF2A, and eIF5B[33, 58, 63–65]. Such factors can promote tRNAᵢ^Met recruitment to the ribosome, as well as allow for initiation with elongator tRNAs, such as leucine-tRNA at CUG codons, when functional eIF2 is limited[33, 58, 63–66]. Empirically identifying the specific tRNA and initiation factors required for RAN initiation at G₄C₂ and CGG repeats will be important moving forward.

In sum, our findings create a framework for better understanding how C9RAN translation occurs mechanistically, while also providing a potential explanation for how such an inefficient form of protein translational initiation can contribute to neurodegeneration. By identifying a central cellular pathway (eIF2α phosphorylation) as a trigger for selective enhancement of RAN translation, we are now well-positioned to explore how both exogenous and endogenous cellular stressors, including repetitive RNAs and RAN translation products themselves, contribute to neurodegeneration. When coupled with the inherent toxicity of RAN-derived proteins and their resistance to degradation, this mechanism creates a mutually reinforcing system that feeds forward to enhance RAN translation and its toxic downstream consequences (Fig. 7)[22, 24, 48]. Interventions which selectively intercede in this feed-forward loop are thus promising targets for future therapeutic development.

## Methods

**Antibodies**. The following antibodies were used for western blots as specified; 1:1000 FLAG-M2 (mouse, Sigma F1804), 1:1000 GAPDH 65C (mouse, Santa Cruz sc32233), 1:1000 tubulin (mouse, DSHB 12G10), 1:1000 PERK (rabbit, CST 3192S), 1:1000 phospho-eIF2α (rabbit, Thermo MA5-15133), 1:1000 BiP (rabbit, CST 3177S), 1:5000 puromycin 12D10 (mouse, Millipore MABE434), in 5% non-fat dry milk (NFDM). LI-COR IRDye 680RD goat-anti-mouse secondary antibody (96-68070) was used 1:10,000 in 5% NFDM for GAPDH and tubulin loading controls. HRP-conjugated goat-anti-mouse (115-035-146) or goat-anti-rabbit (111-035-144) antibodies from Jackson ImmunoResearch Laboratories were used at 1:10,000 in 5% NFDM for all other western blots. Full-length images of the western blots from Figs. 1b and 3b, f are supplied in Supplementary Fig. 7.

The following antibodies were used for immunocytochemistry as specified; 1:500 FLAG (rabbit, Cell Signaling #2368), 1:200 G3BP (mouse, BD Transduction Laboratories 23/G3BP), 1:200 FMRP (mouse, Covance 6B8), and 1:200 FMRP (rabbit, abcam17722) in 5% normal goat serum (NGS—HEK293 cells) or 2% bovine serum albumin (BSA—MEFs). Secondary goat-anti-mouse Alexa Fluor 488 (A-11029) and goat-anti-rabbit Alexa Fluor 555 (A-21428) from Life Technologies were applied at 1:500.

**Plasmids**. For C9RAN reporters, the 5′ end of *C9rof72* intron1 was PCR-amplified from human fibroblast DNA and inserted upstream of previously published GGG-NL-3xF in pcDNA3.1(+)[36] via NheI. Native intronic near-cognate codons were mutated using Q5 Site-Directed Mutagenesis (SDM) Kit (NEB). An AUG start codon was then added to the intronic sequence through a similar strategy. Annealed primers containing PSP cleavage sequence were ligated into an engineered AgeI site upstream of GGG-NL-3xF sequence. Q5 SDM was used to add one or two nucleotides immediately 5′ to the PSP site, to generate reporters for all three sense reading frames, and remove 3′ AgeI site resulting from PSP insertion. 70 $G_4C_2$ repeats were transferred from a published construct[67] immediately 3′ to the intronic sequence and 5′ to the PSP site, via engineered EagI and AscI sites. Repeat sequence contains a single C to A mutation resulting in an imperfect GGGGCA at repeat 13.

Near-cognate NLuc reporters were constructed by mutating the start codon of pcDNA3.1(+)/AUG-NLuc-3xF[36] using the Q5 SDM. pcDNA3.1(+)/ATF4 5′ leader-NLuc-3xF was constructed by subcloning a synthetic insert (Integrated DNA Technologies) into pcDNA3.1(+) via SacI/XbaI. This reporter was designed as previously published for a *ATF4* 5′ leader-FLuc reporter[33], which harbors the complete 5′ leader of the human *ATF4* including the annotated AUG start of ATF4 and the complete overlapping inhibitory uORF.

See Supplementary Tables 1 and 2 for primer and C9RAN NLuc reporter sequences.

**RNA synthesis**. RNAs were in vitro transcribed from linearized plasmids[36]. pcDNA3.1(+) reporter plasmids were linearized with PspOMI; pCRII FLuc reporter with HindIII-HF. Linearized DNA was in vitro transcribed using HiScribe T7 High Yield RNA Synthesis Kit (NEB), with 3′-O-Me-m$^7$GpppG anti-reverse cap analog (ARCA) or ApppG cap (NEB) added at eight times the concentration of GTP, for a capping efficiency of ~90%. 10 µL T7 reactions were carried out at 37 °C for 2 h. Reactions were then treated with 2 U RNase-free DNaseI (NEB) for 15 min at 37 °C to remove DNA template, and then polyadenylated with 5 U *E. coli* Poly-A Polymerase, 10× buffer, and 10 mM ATP (NEB) for 1 h at 37 °C. Synthesized mRNAs were clean and concentrated with RNA Clean and Concentrator-25 Kit from Zymo Research. The size and quality of all synthesized mRNAs were verified on a denaturing formaldehyde RNA gel.

**Rabbit reticulocyte lysate in vitro translation**. mRNAs were in vitro translated with Flexi Rabbit Reticulocyte Lysate System from Promega, that is supplemented with calf liver tRNA[36]. Reactions for luminescence assays were programmed with 3 nM mRNA and contained 30% RRL, 10 mM amino-acid mix minus methionine, 10 mM amino acid mix minus leucine, 0.5 mM MgOAc, 100 mM KCl, and 0.8 U µL$^{-1}$ Murine RNASe Inhibitor (NEB), and incubated at 30 °C for 30 min before termination by incubation at 4 °C. Reactions were then diluted 1:7 in Glo Lysis Buffer (Promega), and incubated 1:1 for 5 min in the dark in opaque 96-well plates with NanoGlo Substrate freshly diluted 1:50 in NanoGlo Buffer (Promega). Luminescence was measured on a GloMax 96 Microplate Luminometer.

For comparison of translation levels between m$^7$G- and A-capped reporters, seven-molar excess of m$^7$G-capped and polyadenylated FLuc mRNA was added to reactions as this has been shown to better recapitulate the endogenous cap and poly (A) synergy[68]. For eIF4E competition assays, 250 µM free ARCA (m$^7$G-cap) or A-cap was added to reaction mixture. For eIF4A inhibition, RRL mix was pre-incubated with 4 µM hippuristanol (a kind gift from Jerry Pelletier, McGill University), prior to addition of NLuc reporters and seven-molar excess FLuc mRNA.

Reactions for western blot assays were performed as above, except 50 ng mRNA was used. 10 µL reactions were mixed with 40 µL sample buffer and heated at 70 °C for 15 min, and 20 µL was run on a 12% polyacrylamide gel.

For precision protease (PSP) site cleavage, 4 µL RRL reaction was mixed either with 17.78 µM cycloheximide, 4 µL RNase-free water, and either 2 U PSP (GE Health Sciences) or vehicle, and incubated for 30 min at 30 °C, prior to processing for luminescence or western blot analysis.

**Transfections and drug treatments**. HEK293, HEK293T, and HeLa cells were purchased from American Type Culture Collection (ATCC). WT and A/A MEFs were received from Randal Kaufman (Sanford Burnham Prebys Medical Discovery Institute).

For C9RAN luminescence assays, HEK293 cells were seeded in 96-well plates at $2 \times 10^4$ cells per well and transfected 24 h later at ~80% confluency. RNA transfections were performed with *TransIT*-mRNA Transfection Kit from Mirus, per manufacturer's recommended protocol, with 90 ng reporter mRNA and 200 ng pGL4.13 FLuc control DNA added to each well in triplicate. C9RAN DNA transfections were performed in triplicate with FuGene HD at a 3:1 ratio to DNA, except where specified below, with 50 ng NLuc reporter DNA and 50 ng pGL4.13 FLuc reporter added per well. Cells were then lysed 24 h post transfection with 60 µL Glo Lysis Buffer for 5 min at room temperature. A concentration of 25 µL of lysate was mixed with NanoGlo Substrate prepared as for RRL reactions, and 25 µL of ONE-Glo Luciferase Assay System (Promega), for 5 min in the dark, in opaque 96-well plates. Luminescence measurements were obtained as with RRL reactions. *P* values were calculated using Student's *t* test.

For C9RAN reporter luminescence analysis following ISR activation, HEK293 cells were seeded and transfected as above for 19 h, followed by 5 h drug treatment. For CGG RAN reporter luminescence analysis, HEK293T cells were seeded at $1.5 \times 10^4$ cells per 96 well for 36 h, then transfected with 4:1 Viafect:DNA for 1 h, followed by 5 h drug treatment. MEFs were seeded at $1 \times 10^4$ cells per well for 24 h, then transfected 2:1 with jetPRIME:DNA for 1 h, followed by a 5 h drug treatment. All cell types were lysed and luciferase activity measured as above. Drugs used: TG (Thermo), Sal003 (Sigma), TM (Sigma), and SA (Sigma).

For assessment of RAN translation in the presence of overexpressed eIF2α-S51D, HEK293 and 293T cells were grown as above. NLuc reporters were co-transfected 1:1 with pGL4.13 FLuc, and 1:10 with an effector plasmid (eIF2α WT or S51D) for 24 h. Cells were lysed and luciferase activity was measured as above.

For C9RAN western blots, HEK293 cells were seeded in 12-well plates at $2 \times 10^5$ cells per well and transfected 24 h later at ~80% confluency with 500 ng NLuc reporter DNAs and 4:1 FuGene HD. Cells were lysed in 300 µL RIPA buffer with protease inhibitor 24 h post transfection, for 30 min at 4 °C. Lysates were homogenized by passing through a 28G syringe, mixed with 6X sample buffer, and stored at −20 °C.

For western analysis of RAN reporters following stress induction, HEK293T cells were seeded at $7.5 \times 10^4$ cells per well in 24-well plates. Twenty-four hours post seeding, cells were transfected with 250 ng NLuc reporter plasmids and 250 ng pGL4.13 FLuc plasmids with 3:1 FuGene HD for 18 h, followed by 5 h drug treatment. Cells were lysed as with C9RAN western blot transfections. Each genotype was run in triplicate on a 12% SDS-PAGE along with a standard curve for quantification of protein expression. Band intensities were measured using ImageJ, and quantified by extrapolating off the standard curve, and normalizing to alpha-tubulin.

**HeLa cell lysate in vitro translation**. In vitro translation extracts were prepared from cultured HeLa cells[37] (ATCC) maintained at 37 °C, 5% $CO_2$ in DMEM supplemented with 10% FBS and 1% non-essential amino acids. To prepare extracts, adherent cells were trypsinized, centrifuged, and washed in PBS. Cell pellets were resuspended in RNAse-free hypotonic buffer containing 10 mM HEPES-KOH (pH 7.6), 10 mM potassium acetate, 0.5 mM magnesium acetate, 5 mM DTT, and EDTA-free protease inhibitor cocktail (Roche). Cell pellets were incubated on ice for 20 min, mechanically disrupted by a 27G syringe, incubated for another 20 min on ice, and centrifuged at $10,000 \times g$ for 10 min at 4 °C. The supernatant was removed, then brought to 4 µg µL$^{-1}$ in additional hypotonic buffer. For in vitro translation reactions, lysates were supplemented to final concentrations of 20 mM HEPES-KOH (pH 7.6), 44 mM potassium acetate, 2.2 mM magnesium acetate, 2 mM DTT, 20 mM creatine phosphate (Roche), 0.1 µg µl$^{-1}$ creatine kinase (Roche), 0.1 mM spermidine, and on average 0.1 mM of each amino acid (with relative amounts approximating those in eukaryotes[69]). To this, in vitro transcribed reporter RNAs were added to 4 nM. After incubation at 30 °C for 30 min, luciferase assays were carried out as with RRL reactions.

**Primary rat hippocampal neuron transfection**. Rat hippocampi were collected from postnatal day 0–2 pups, dissociated with L-cysteine-activated papain, and 60,000 neurons were plated per well on poly-D-lysine coated coverslips in neuronal growth media (NGM). Neurons were allowed to mature for 13 days in vitro, with half NGM media changes, supplemented with glial and cortical enriched media, every 2–3 days. On DIV13, neurons were transfected with 5 µg DNA and 10 µL Lipo2000 per well. 48 h post transfection, neurons were lysed in 300 µL Glo Lysis Buffer for 5 min at room temperature. 80 µL of lysed cells were incubated with 80 µL freshly prepared NanoGlo Substrate in NanoGlo Buffer or ONE-Glo, and luminescence measured as with other assays.

**Protein stability analysis**. Twenty-four hours post RAN plasmids transfection, performed as above, HEK293 cells were treated with 10 µg ml$^{-1}$ puromycin for 0, 6, and 24 h. After each timepoint, cells were lysed in 60 µL Glo Lysis Buffer for 5 min at room temperature and stored at −20 °C. After all time points were collected, NLuc and FLuc activities was measured simultaneously.

**Automated fluorescence microscopy imaging of primary neurons**. Rat cortical primary neurons were harvested from E20 pups and cultured at $0.6 \times 10^6$ cells per mL in vitro. On DIV4, neurons were co-transfected with 0.1 µg pGW1-GFP, pGW1-($G_4C_2$)×66-GFP, or pGW1-FMRP-(CGG)x100-GFP DNA and 0.1 µg pGW1-mApple with 2:1 Lipo2000 (Invitrogen, 52887). Beginning 1 day post transfection, neurons were reiteratively imaged with automated fluorescent microscopy for four to 10 days[70, 71]. Image processing and fluorescent intensity measurements for GFP and mApple ($n > 30$ neurons) were obtained for each timepoint using custom code written in Python or the ImageJ macro language. To assess the effects on ISR activation on RAN translation in neurons, cells with treated with 0.5, 1, or 2 µM TG following the first timepoint.

**Monitoring translation by puromycin incorporation**. Translation levels were assessed using the surface sensing of translation (SUnSET) method[50]. HEK293 cells were seeded at $1 \times 10^5$ cells per well in 24-well plates, and transfected 24 h later with 250 ng CGG RAN reporters and 4:1 FuGene HD. 24 h after transfection, cells

were incubated with fresh media containing 10 µg ml$^{-1}$ puromycin for 10 min at room temperature. Cells were then placed on ice and washed with ice-cold PBS, prior to lysis in 150 µL RIPA buffer containing protease inhibitor.

**Stress granule analysis**. HEK293 cells were seeded at $1 \times 10^5$ cells per well in 4-well chamber slides 24 h prior to FuGene HD transfection of 250 ng DNA reporters. Twenty-four hours post transfection, cells were fixed with 4% paraformaldehyde (PFA) in PBS-MC for 15 min at room temperature, permeabilized with 0.1% triton-X in PBS-MC for 5 min at room temperature, blocked with 5% NGS, and incubated overnight with primary antibodies in 5% NGS at 4 °C in a humidity chamber. The following morning, cells were incubated with Alexa-Fluor secondary antibodies for 1 h at room temperature in the dark. Coverslips were then applied to slides with ProLong Gold Antifade Mountant with DAPI. 3–5 fields per condition were imaged at 20 × 1.6 magnification with Olympus IX71 fluorescent microscope and Slidebook 5.5 software.

WT and A/A mutant MEFs were seeded at $1 \times 10^5$ cells per well for 24 h, then transfected with 500 ng NLuc reporters and 2:1 jetPRIME for 24 h. Cells were fixed and permeabilized as above, blocked with 2% BSA for 20 min at room temperature, and incubated overnight with primary antibodies in 2% BSA at 4 °C. Secondary antibodies were applied the following morning for 1 h at room temperature, in the dark. A total of 10–20 fields per condition were taken at 20 × 1.6 magnification, as above.

For stress granule analysis, signals for each channel were normalized prior to quantification. For HEK239 cells, >450 cells were counted for each genotype (>70 transfected cells/genotype). For MEFs, >370 cells were counted for each genotype (>40 transfected cells/genotype). Quantification was performed using ImageJ analysis. P values were calculated using Fisher's exact test.

**Statistical methods**. Statistical analysis was performed using GraphPad Prism7. For comparison of NLuc reporter luciferase activity, one-way ANOVAs were performed to confirm statistical difference between control and experimental groups. Post-hoc Student's $t$ tests were then performed with Bonferroni correction for multiple comparisons and Welch's correction for unequal variance. Fisher's exact tests were used for immunocytochemistry experiments, to determine if there was a statistical difference between the proportion of control or RAN transfected cells that contained stress granules.

**Data availability**. The data that support the findings of this study are available from the corresponding author upon request.

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

## Acknowledgements

We thank members of the Todd lab for input on this manuscript. Jerry Pelletier (McGill University) kindly provided hippuristanol, and Randal Kaufman (Sanford Burnham Prebys Medical Discovery Institute) supplied WT and eIF2α S51 A/A MEFs. This work was funded by grants from the VA BLRD (1I21BX001841 and 1I01BX003231), the NIH (R01NS099280 and R01NS086810), the Packard Foundation, the Michigan Alzheimer's Disease Center and Protein Folding Disease Initiative to P.K.T. K.M.G., B.N.F., and A.E. L. were supported by NIH T32GM007315. K.M.G. was further supported by NIH F31NS100302 and A.E.L. by NIH F30NS098571. M.R.G. was supported by NIH T32NS007222-35S1. M.G.K. was supported by the NIH F32NS089124. S.J.B. and B.N.F. were supported by the NIH (R01-NS097542, 1P30AG053760-01) and Ann Arbor Active Against A.L.S.

## Author contributions

K.M.G., M.R.G., M.G.K., A.C.G., P.K.T.: Planned and conceived the experiments.
K.M.G., M.R.G., M.G.K.: Performed the experiments with help from A.E.L. and S.J.F.
B.N.F., and S.J.B.: Performed the automated microscopy experiments in primary neurons.
K.M.G., M.R.G., M.G.K., and P.K.T.: Wrote the paper with input from all authors.

## Additional information

**Competing interests:** P.K.T. serves as a consultant with Denali Therapeutics and has licensed technology through the University of Michigan to Denali that is based on work published here. The remaining authors declare no competing financial interests.

