## [Peer Review File · Nature Communications]

Reviewers' comments:

Reviewer #1 (Remarks to the Author):

In this study, Green et al. examined unconventional (RAN) translation at G4C2 and CGG repeat expansions that causes C9orf72-associated neurodegenerative disorders (ALS, FTD and FXTAS). They concluded that the C9RAN translation is initiated at a CUG codon by a cap-dependent scanning mechanism. RAN translation at both CGG and G4C2 repeats is enhanced under stress conditions, and the fidelity of start codon selection depends on eIF2alpha phosphorylation. In addition, the CGG and G4C2 repeats induce stress granule formation in eIF2alpha phosphorylation dependent manner.

This is an interesting work that advances the understanding of the mechanism of RAN translation at the G4C2 and CGG repeat expansions. The demonstrated ability of these repeats to induce eIF2alpha phosphorylation and the potentiation of RAN translation by this process suggest the existence of a feed-forward loop that contributes to the accumulation of toxic proteins, a hallmark of many neurodegenerative disorders. There are some issues that should be addressed before publication.

The current study along with previous work by the same group on FMR1 (Kearse et al. 2016), suggest that regardless of the sequence and length (Figure 1b) of the repeats the mechanism of RAN translation remained identical. This in vitro observation goes against the pathophysiology of repeat-associated diseases where the sequence and the length of repeats determine disease severity. Along these lines, replacing non-AUG start codon with AUG in Figure 5 completely blunts the activating affect of TG and Sal003 on RAN translation and this effect is independent of CGG or G4C2 repeats.

Comments:

1. Fig.1: Would synthesis of the GA, GP and GR DPRs from the C9RAN mRNA raise/persist after substitution of AUG for CUG?
2. Fig.2c shows only modest (~2.5-fold) inhibition of the control AUG mRNA by the m7GpppG cap analogue, indicating the relaxed cap-dependence of their system. However, in contrast, there is a huge translational difference between m7G- and A- capped control mRNAs. Why? Can this

be explained?

3. Fig.2e: Loss of upstream near-cognate CUG start codon does not cause a reduction, but instead significantly enhances RAN translation in the GP reading frame in RRL. If CUG is not used to initiate synthesis of GP DPR, then how is initiation in the GP frame occurring?
4. Fig.3c and 3e: There is almost no inhibition of the AUG-NLuc reporter by TG (in contrast to their statement on lines 233-235). This is surprising, since TG markedly inhibits FLuc mRNA (Figs.3c and 3e) and general translation (Supplementary Fig.3a and Fig.5b).
5. Fig.4 and/or Supplementary Fig.3: The effects of Sal003 and TM on eIF2alpha phosphorylation in HEK293 cells should be shown.
6. Fig.5c shows that near-cognate codons are also required for stress-induced conventional translation. Therefore, the title for Fig.5 should be altered to read "...for stress-induced translation".
7. Page 4 line 89: "eIF2 then hydrolyzes GTP to GDP and is released..." should be replaced by "eIF5 then hydrolyzes GTP to GDP and eIF2-GDP is released..."

Reviewer #2 (Remarks to the Author):

This manuscript addresses several issues associated with RNA translation, the start codon, the mechanism for finding the start codon and the perhaps surprising result of up-regulation of RAN translation under conditions that trigger eIF2alpha phosphorylation (via the integrated stress response). And potentially a major concern tied to this is that the dipeptide repeats associated with microsatellite repeat diseases may even enhance the stress signal and thereby further amplify RAN translation. This manuscript represents a nice start at beginning to unravel this mystery although there is clearly room for further clarification, most notably, what protein is responsible for directing the binding of the initiating aminoacyl-tRNA.

Specific concerns

1. In Figure 1, panel C, the yield of the three different dipeptide repeats is given with GA>GP>GR. The question is, does this represent the tRNA populations present in RRL? An examination of the rabbit beta globin chain mRNA indicates the following uses of codons that might arise from the G4C2 repeat: arginine – no CGG codons used; alanine – half of the codons used are GCC; proline – no CCG codons used; glycine – more than half of the codons used are either GGG or GGC. If one assumes that in this cell which will be almost 95-98% hemoglobin that the tRNA population is a match to the amino acids in hemoglobin, might this then be reflected in the synthesis seen in panel C? And might this also account for the differences seen in panels D and E which are very similar, but dissimilar to panel C?

2. From Figure 2, it is concluded that RAN translation occurs by a scanning mechanism because its synthesis is inhibited by the eIF4A inhibitor hippuristanol. However, there are two steps in translation that require eIF4A, mRNA activation and scanning. The affect of hippuristanol could be on either of these two steps. Therefore, this drug inhibition is not proof of scanning.
3. The alternate use of CUG codons to initiate has been noticed recently (see Stark et al. Science, 2016; Sendoel et al. Nature 2017). In RRL, 15 of the 17 leucine codons are CUG and thus leucyl-tRNA would be an excellent candidate for the initiator tRNA. As above, is the effectiveness of initiator tRNA in RAN translation influenced by aminoacyl-tRNA availability?
4. While the loss of ternary complexes may favor alternate initiation schemes, the other part of translation is the selection of mRNA which is influenced by the mTOR pathway. In the cells that are stressed (yielding increased eIF2alpha phosphorylation), is there any change in the level of phosphorylation of 4E-BP which might then favor utilization of mRNAs that are not the usual m7G cap-dependent mRNAs.
5. On line 272 it states: “Increased eIF2alpha phosphorylation is necessary for stress-induced RAN translation.” However, the stimulation by Sal003 appears to be rather modest, 30% (Figure 4, panel A). It would be more appropriate to indicate that the phosphorylation “enhances” stress-induced RAN translation.
6. Line 430 – “...by creating a blockade for scanning 40S ribosomes...” Might this be similar to initiation of protein synthesis for alphaviruses (Ventoso et al. Genes Dev 2006)?
7. In either cell free systems or in vivo, it would be anticipated that a capped mRNA would be translated more efficiently. What is the evidence that the in vivo GGGGCC repeat RNA is capped (and also perhaps, polyadenylated)?

Minor concerns

1. Might the authors consider using the inhibitors NSC119893 (block eIF2) and arcifavin (block CUG utilization) as was done by Starck et al. Science 2012?
2. In these studies, it does need to be emphasized that RAN synthesis is very inefficient compared to normal protein synthesis. Given the progressive nature of ALS, the question is raised as to whether this is mostly a disease where the dipeptide repeat proteins just don't turn over (i.e. have a very extended lifetime) but may accelerate when the level of this proteins reaches a certain threshold?
3. The authors need to be more consistent in their use of time abbreviations, especially in Methods.
4. Figure 2 or Methods – What is the concentration of the inhibitor used (ApppG and m7GpppG)? Here and elsewhere, a bit more information in the figure legends would be helpful.

Responses to specific Reviewers' comments below (reviewer comments in italics):

Reviewer #1:

In this study, Green et al. examined unconventional (RAN) translation at G4C2 and CGG repeat expansions that causes C9orf72-associated neurodegenerative disorders (ALS, FTD and FXTAS). They concluded that the C9RAN translation is initiated at a CUG codon by a cap-dependent scanning mechanism. RAN translation at both CGG and G4C2 repeats is enhanced under stress conditions, and the fidelity of start codon selection depends on eIF2alpha phosphorylation. In addition, the CGG and G4C2 repeats induce stress granule formation in eIF2alpha phosphorylation dependent manner.

The current study along with previous work by the same group on FMRI (Kearse et al. 2016), suggest that regardless of the sequence and length (Figure 1b) of the repeats the mechanism of RAN translation remained identical. This in vitro observation goes against the pathophysiology of repeat-associated diseases where the sequence and the length of repeats determine disease severity. Along these lines, replacing non-AUG start codon with AUG in Figure 5 completely blunts the activating effect of TG and Sal003 on RAN translation and this effect is independent of CGG or G4C2 repeats.

This is an interesting work that advances the understanding of the mechanism of RAN translation at the G4C2 and CGG repeat expansions. The demonstrated ability of these repeats to induce eIF2alpha phosphorylation and the potentiation of RAN translation by this process suggest the existence of a feed-forward loop that contributes to the accumulation of toxic proteins, a hallmark of many neurodegenerative disorders. There are some issues that should be addressed before publication.

Comments:

1. Fig.1: Would synthesis of the GA, GP and GR DPRs from the C9RAN mRNA raise/persist after substitution of AUG for CUG?

To determine the relative differences in initiation between AUG-initiated translation through the repeat and initiation from the CUG codon, we placed an AUG codon in a strong Kozak context upstream of the repeat in the GA reading frame. This leads to a significant enhancement (30 fold) in NLuc signal in HEK293 cells compared to the native intronic sequence alone. These same experiments in RRL show the same trend though with somewhat reduced magnitude (6 fold). This data is now included in Supplementary Fig. 2e, g.

A single CUG codon in the GA reading frame appears to be critical for translation in the GA reading frame but also influences synthesis in the GR and GP reading frames, with enhanced synthesis of GP DPRs and decreased synthesis of GR DPRs when this CUG is mutated to CCC in RRL, although these effects are less clear in transfected cells (Figure 2e and Supplementary Figure 2b). In this context, it is reasonable to ask whether GR and GP synthesis demonstrates a reciprocal relationship when this CUG codon is changed to an AUG codon. To determine this, we mutated this CUG codon to AUG. In RRL,

this leads to a significant decrease in GP translation but an increase in GR synthesis. However, in transfected cells, changing this codon to AUG suppressed production in both the GP and GR reading frames. Overall, these findings suggest that translation in RRL exhibits a pattern consistent with scanning-mediated initiation at the CUG codon and competition between the GA and GP reading frames, while effects on GR production suggest a possible frameshift from the GA to GR reading frame. In transfected cells, however, our findings are more consistent with a direct competition for initiation between all three reading frames without evidence suggestive of frameshifting.

This data is now included in Supplementary Fig. 2 and shown below.

New Supplementary Fig. 2d, f: (d) Expression of GP and GR-NLuc reporters in RRL from constructs with CUG codon mutated to AUG, relative to WT sequence in RRL, n=6. (f) Mutating CUG codon to AUG decreases expression of GP and GR-NLuc reporters in HEK293 cells, n=6. Two-tailed student's t test with Welch's correction, ****p<0.0001.

2. Fig.2c shows only modest (~2.5-fold) inhibition of the control AUG mRNA by the m⁷GpppG cap analogue, indicating the relaxed cap-dependence of their system. However, in contrast, there is a huge translational difference between m⁷G- and A- capped control mRNAs. Why? Can this be explained?

This difference is likely explained by previous work from the Merrick and Jankowsky groups on eIF4F RNA binding. They demonstrated that eIF4F binds much more avidly to capped RNA than to cap analog, because eIF4F has greater contact throughout the RNA rather than just the site of binding to the cap¹. Thus, competitive inhibition by cap analog is not as effective at titrating the eIF4F complex off the capped mRNA.

We have previously tested different concentrations of free m⁷G cap in the RRL reaction on AUG-NLuc production (Kearse et al, Mol Cell, 2016). At higher doses (500μM), there is impairment of IRES-based translation from our CrPV reporter. As the cap analog is negatively charged, these higher concentrations are probably sequestering Mg²⁺. Multiple reports show how altering Mg²⁺ concentrations can inhibit/affect *in vitro* translation reactions (please see Promega product reference sheet related to the RRL system (<https://www.promega.com/-/media/files/resources/promega-notes/74/ribo-m7g-cap-analog-a-reagent-for-preparing-in-vitro-capped-transcripts.pdf?la=en>) and ^{2,3}). In agreement with this, we and others⁴ have found that KCl supplementation, and not KOAc, gives far better cap-dependency in RRLs.

3. Fig.2e: Loss of upstream near-cognate CUG start codon does not cause a reduction, but instead significantly enhances RAN translation in the GP reading frame in RRL. If CUG is not used to initiate synthesis of GP DPR, then how is initiation in the GP frame occurring?

Our current data are most consistent with a model where synthesis in the GP reading frame initiates within the repeat itself at a non-AUG codon. This is based partially on the native sequence; the last codon before the repeat in the GP reading frame is a UAG stop codon. Thus, initiation 5' to this repeat would not translate a product through the repeat due to this stop codon unless there was read-through. How exactly initiation occurs within the repeat and what codon is used remains unclear but is an area of ongoing research. Work on CAG repeats suggests a similar initiation for generation of the polyalanine product from the GCA codon, with the initial amino acid appearing to be alanine rather than methionine (Zu et al, PNAS, 2011).

The CUG start codon is located in the GA reading frame. If C9RAN translation utilizes a scanning mechanism of initiation, then a scanning 43S Preinitiation complex (PIC) would reach the CUG codon prior to proceeding into the repeat itself. If this CUG codon is removed, scanning PICs would initiate less readily upstream of the repeat, leading to more scanning PICs that could potentially initiate within the repeat. Thus, these repeat reading frames appear to “compete” with one another for scanning PICs, with usage favored towards the 5' CUG codon. We feel that these data support a scanning model of initiation for C9RAN translation. We now explain our interpretation of this finding more clearly in the discussion of the paper.

4. Fig.3c and 3e: There is almost no inhibition of the AUG-NLuc reporter by TG (in contrast to their statement on lines 233-235). This is surprising, since TG markedly inhibits FLuc mRNA (Figs.3c and 3e) and general translation (Supplementary Fig.3a and Fig.5b).

This effect is likely due to the stability of AUG-NLuc, which is greater than both the FLuc and the RAN-NLuc reporters over the time course utilized for these studies (See Kearse et al, 2016 and Supplementary Fig. 1d). Consistent with this prediction, treatment of cells transfected with a modified version of AUG-NLuc that has a PEST tag (which causes faster protein turnover) at its C-terminus exhibits lower expression after treatment with thapsigargin. These data are now included below and in Supplementary Fig. 3b.

5. Fig.4 and/or Supplementary Fig.3: The effects of Sal003 and TM on eIF2alpha phosphorylation in HEK293 cells should be shown.

We now include these controls in Supplementary Fig. 4a and below. As expected, they demonstrate increased phosphorylation of eIF2alpha at doses and durations utilized for our RAN translation assays.

New Supplementary Fig. 4a: Western blot showing increased phosphorylation of eIF2 α in HEK293 cells following treatment with Sal003 (20 or 40 μ M), TM (1 or 2.5 μ g/mL), and SA (20 μ M).

6. Fig.5c shows that near-cognate codons are also required for stress-induced conventional translation. Therefore, the title for Fig.5 should be altered to read “...for stress-induced translation”.

Given that there are numerous examples in the literature for altered use of AUG-initiated uORFs that allow for translation after activation of the integrated stress response (e.g. See Hinnenbush et al, Science, 2016), it would be inaccurate to state that near-cognate codons are required for stress-induced translation. We have therefore changed the title for Figure 5 to “**Near-cognate codons are sufficient to allow for stress-induced translation**”.

7. Page 4 line 89: “eIF2 then hydrolyzes GTP to GDP and is released...” should be replaced by “eIF5 then hydrolyzes GTP to GDP and eIF2-GDP is released...”

eIF5 is the eIF2-specific GTPase-activating protein. It promotes GTP hydrolysis by eIF2 but does not itself directly bind or hydrolyze GTP. We have therefore reworded this sentence as follows: “**eIF5 then promotes hydrolysis of GTP to GDP on eIF2, and eIF2-GDP and P_i is released...**”

Reviewer #2:

This manuscript addresses several issues associated with RNA translation, the start codon, the mechanism for finding the start codon and the perhaps surprising result of up-regulation of RAN translation under conditions that trigger eIF2alpha phosphorylation (via the integrated stress response). And potentially a major concern tied to this is that the dipeptide repeats associated with microsatellite repeat diseases may even enhance the stress signal and thereby further amplify RAN translation. This manuscript represents a nice start at beginning to unravel this mystery although there is clearly room for further clarification, most notably, what protein is responsible for directing the binding of the initiating aminoacyl-tRNA.

Specific concerns

1. In Figure 1, panel C, the yield of the three different dipeptide repeats is given with GA>GP>GR. The question is, does this represent the tRNA populations present in RRL? An examination of the rabbit beta globin chain mRNA indicates the following uses of codons that might arise from the G4C2 repeat: arginine – no CGG codons used; alanine – half of the codons used are GCC; proline – no CCG codons used; glycine – more than half of the codons used are either GGG or GGC. If one assumes that in this cell which will be almost 95-98% hemoglobin that the tRNA population is a match to the amino acids in hemoglobin, might this then be reflected in the synthesis seen in panel C? And might this also account for the differences seen in panels D and E which are very similar, but dissimilar to panel C?

This is an interesting idea. We are unaware of evidence demonstrating that there is a skewed population of tRNAs in rabbit reticulocyte lysates. However, even if this were the case, our RRL reactions are supplemented by the manufacturer with 50 μ g/ml of calf liver tRNA. Thus, there is unlikely to be a

specific shortage or excess of any tRNA in the *in vitro* reaction that would strongly influence translational efficiency.

This reviewer concern, however, raises a broader question; are the differences observed in translation across the different G₄C₂ repeat frames due to differences in the rates of initiation or elongation? And, as a corollary, is there something about the RRL (with differential tRNA abundance as one feature) that leads to more pronounced differences between reading frames compared to experiments performed in cells or neurons?

To test these possibilities, we have performed two new experiments.

a) We have introduced an AUG codon in strong Kozak context into the RNA 5' to the repeats in all three potential reading frames. For the GP reading frame, we have also removed the stop codon that normally exists just 5' to the repeat sequence. These constructs should thus provide the same initiation sites in the same sequence context, and we would expect the rates of initiation to be comparable between them, allowing them to serve as reporters for differential elongation. In RRL and cells, we find that the expression of AUG-GA70 and AUG-GP70 are comparable. However, in both contexts, AUG-GR70 expression is lower. Of note, however, this decrease is much less than that seen between GA70 and GR70 *in vitro*, suggesting that there are differences in both initiation and elongation rates between these reading frames. These data are now included in Supplementary Fig. 1g and h and shown below.

New data from Supplementary Fig. 1g, h: (g) Expression from AUG-driven reporters for each sense reading frame, relative to AUG-GA70, in RRL, n=6 and (h) HEK293 cells, n=6. Two-tailed student's t test with Welch's correction, NS- not significant; ***p<0.0001; ****p<0.0001.

b) To determine whether *in vitro* translation in a different system alters the relationship across reading frames, we utilized in-house-prepared HeLa cell lysate from published protocols⁵. These have been used previously by other groups with good success to study aspects of translational initiation. Using this system, we compared the relative translation readouts across reading frames. As with the RRL and transfected cells, we observe that RAN translation is less efficient than AUG initiated translation overall. When comparing reading frames, we again observe that products generated from the GA reading frame are significantly more abundant by luciferase assay than those generated from GP and GR70 constructs. These data are shown below and are now included in the revised manuscript as Supplementary Fig. 1c.

2. From Figure 2, it is concluded that RAN translation occurs by a scanning mechanism because its synthesis is inhibited by the eIF4A inhibitor hippuristanol. However, there are two steps in translation that require eIF4A, mRNA activation and scanning. The effect of hippuristanol could be on either of these two steps. Therefore, this drug inhibition is not proof of scanning.

We agree that eIF4A has multiple functions and that its blockade could impede RAN translation by altering ribosomal loading/mRNA activation rather than scanning per se. We now acknowledge this consideration in the discussion and modify our claim related to scanning, saying that our findings are “consistent with a scanning model of initiation.”

However, all of our data to date, including studies not included in this manuscript, strongly support a scanning mechanism. For example, in work that is part of a related study, we find that placing an AUG initiated open reading frame above the intronic sequence dramatically reduces C9RAN translation in all reading frames both in RRL and cells (not shown). Thus, while we acknowledge the shortcoming of this particular finding as non-definitive, in sum the preponderance of evidence supports a scanning mechanism.

3. The alternate use of CUG codons to initiate has been noticed recently (see Stark et al. Science, 2016; Sendoel et al. Nature 2017). In RRL, 15 of the 17 leucine codons are CUG and thus leucyl-tRNA would be an excellent candidate for the initiator tRNA. As above, is the effectiveness of initiator tRNA in RAN translation influenced by aminoacyl-tRNA availability?

As noted above, our RRL reactions are supplemented by the manufacturer with 50µg/ml of calf liver tRNA. Thus, there is unlikely to be a specific shortage or excess of any tRNA in the *in vitro* reaction that would strongly influence translational efficiency.

There is clear evidence that CUG can be initiated with either Met-tRNA_i^{Met} or the Leu-tRNA^{Leu/CUG}, both *in vitro* and *in vivo*, in different contexts⁶⁻⁸. We agree with the reviewer that determination of the initiator tRNA that is utilized in C9RAN translation in each reading frame is an important long-term goal and one we are interested in pursuing. However, as these experiments would require us to develop a system for measuring translation where we could deplete and then resupply specific tRNAs, we feel that it is beyond the scope of this manuscript. We have instead expanded our discussion of how C9RAN translation fits within the context of known alternative initiation mechanisms that cites the papers referenced above, providing insights into future directions that will need to be pursued to further our understanding.

4. *While the loss of ternary complexes may favor alternate initiation schemes, the other part of translation is the selection of mRNA which is influenced by the mTOR pathway. In cells that are stressed (yielding increased eIF2alpha phosphorylation), is there any change in the level of phosphorylation of 4E-BP which might then favor utilization of mRNAs that are not the usual m7G cap-dependent mRNAs.*

We agree that mTOR signaling plays a major role in translational regulation, and there is evidence that Thapsigargin treatment (at longer durations than those used in our assays) increases 4E-BP1 levels⁹. However, our data strongly implicates eIF2alpha phosphorylation as the triggering event in enhanced RAN translation. Direct manipulations of eIF2alpha are sufficient to selectively activate RAN translation across reading frames and repeats, and eIF2alpha phosphorylation is required to observe these enhancements. In addition, we provide evidence that C9RAN translation is strongly cap-dependent both *in vitro* and in cells, making escape from ISR suppression via a cap-independent mechanism unlikely. Thus, we do not think that 4E-BP phosphorylation is likely to play a selective role in RAN translational activation.

5. *On line 272 it states: "Increased eIF2alpha phosphorylation is necessary for stress-induced RAN translation." However, the stimulation by Sal003 appears to be rather modest, 30% (Figure 4, panel A). It would be more appropriate to indicate that the phosphorylation "enhances" stress-induced RAN translation.*

We agree. This has now been changed to **"eIF2alpha phosphorylation selectively enhances RAN translation."**

6. *Line 430 – "...by creating a blockade for scanning 40S ribosomes..." Might this be similar to initiation of protein synthesis for alphaviruses (Ventoso et al. Genes Dev 2006)?*

We agree that these two processes might be related and now include this reference in the discussion.

7. *In either cell free systems or in vivo, it would be anticipated that a capped mRNA would be translated more efficiently. What is the evidence that the in vivo GGGGCC repeat RNA is capped (and also perhaps, polyadenylated)?*

Current evidence suggests that repeat containing introns in C9orf72 mRNAs are retained at an increased rate (Gallo et al, 2016), although this finding is somewhat controversial (e.g. Zu et al, PNAS, 2013). These intron-retaining mRNAs are polyadenylated (RT-PCR studies were performed with oligo-dT primers for the reverse transcription step), suggesting that these mRNAs follow the canonical mode of mRNA transcription by PolII, and are presumably capped. Moreover, it appears that retained intronic mRNAs from cells with expanded repeats escape to the cytoplasm at a greater rate, perhaps due to interactions with SRSF1¹⁰ and that this is important in allowing for efficient RAN translation.

Our own data are complementary to this hypothesis. We find that RAN translation from GGGGCC repeats is largely cap-dependent in mRNA transfected cells, which would predict that translation from uncapped mRNAs would be much less efficient. In addition, uncapped mRNAs would be subject to rapid degradation by exoribonucleases, namely XRN1, and would be unlikely to be transported out of the nucleus. Thus, our current data suggests that capped mRNAs, perhaps from retained introns, would be the most likely templates for RAN translation in patient cells. We now include an expanded discussion of this point and how our work might predict which mRNA transcripts are likely to undergo C9RAN translation from the endogenous locus in patient cells.

Minor concerns

1. Might the authors consider using the inhibitors NSC119893 (block eIF2) and arcifavin (block CUG utilization) as was done by Starck et al. Science 2012?

We agree that our findings are reminiscent of those previously reported for initiation at CUG codons in the context of MHC antigen diversity and we have included the above reference in our manuscript. To evaluate this further, we tested the impact of these two agents on AUG and C9RAN translation reporters. We find that Acriflavine shows a relatively selective inhibition of C9RAN translation in all three reading frames, in compared to AUG-NLuc translation, in our RRL system. This result is shown below. It was also selective for RAN translation in the GA reading frame relative to AUG-NLuc, in HEK293 cells. Unfortunately, this drug was toxic at higher doses in HEK293 cells, and highly toxic in neurons even at low concentrations. NSC119893 inhibited translation of all constructs tested, including CrPV IRES-mediated translation, which should initiate absent the Met-tRNA_i^{Met} or any aminoacyl-tRNA in the P-site of the 40S ribosomal subunit, complicating our interpretation of this compound.

2. In these studies, it does need to be emphasized that RAN synthesis is very inefficient compared to normal protein synthesis. Given the progressive nature of ALS, the question is raised as to whether this is mostly a disease where the dipeptide repeat proteins just don't turn over (i.e. have a very extended lifetime) but may accelerate when the level of this proteins reaches a certain threshold?

This point is valid. We now include a discussion of it within the manuscript.

3. The authors need to be more consistent in their use of time abbreviations, especially in Methods.

We apologize for our lack of clarity. We have now removed all time abbreviations to make sure they are consistent throughout the manuscript.

4. Figure 2 or Methods – What is the concentration of the inhibitor used (ApppG and m7GpppG)? Here and elsewhere, a bit more information in the figure legends would be helpful.

We used this at 250μM and previously performed titration curves to optimize this experiment (Kearse et al, Mol. Cell, 2016). We have now added this information to both the methods and the figure legend.

1. Kaye, N.M., Emmett, K.J., Merrick, W.C. & Jankowsky, E. Intrinsic RNA binding by the eukaryotic initiation factor 4F depends on a minimal RNA length but not on the m7G cap. *The Journal of biological chemistry* **284**, 17742-17750 (2009).
2. Soto Rifo, R., Ricci, E.P., Decimo, D., Moncorge, O. & Ohlmann, T. Back to basics: the untreated rabbit reticulocyte lysate as a competitive system to recapitulate cap/poly(A) synergy and the selective advantage of IRES-driven translation. *Nucleic acids research* **35**, e121 (2007).
3. Snyder, R.D. & Edwards, M.L. Effects of polyamine analogs on the extent and fidelity of in vitro polypeptide synthesis. *Biochemical and biophysical research communications* **176**, 1383-1392 (1991).
4. Jackson, R.J. Potassium salts influence the fidelity of mRNA translation initiation in rabbit reticulocyte lysates: unique features of encephalomyocarditis virus RNA translation. *Biochimica et biophysica acta* **1088**, 345-358 (1991).
5. Rakotondrafara, A.M. & Hentze, M.W. An efficient factor-depleted mammalian in vitro translation system. *Nature protocols* **6**, 563-571 (2011).
6. Starck, S.R. *et al.* A distinct translation initiation mechanism generates cryptic peptides for immune surveillance. *PLoS one* **3**, e3460 (2008).
7. Starck, S.R. *et al.* Leucine-tRNA initiates at CUG start codons for protein synthesis and presentation by MHC class I. *Science* **336**, 1719-1723 (2012).
8. Peabody, D.S. Translation initiation at non-AUG triplets in mammalian cells. *The Journal of biological chemistry* **264**, 5031-5035 (1989).
9. Yamaguchi, S. *et al.* ATF4-mediated induction of 4E-BP1 contributes to pancreatic beta cell survival under endoplasmic reticulum stress. *Cell metabolism* **7**, 269-276 (2008).
10. Hautbergue, G.M. *et al.* SRSF1-dependent nuclear export inhibition of C9ORF72 repeat transcripts prevents neurodegeneration and associated motor deficits. *Nature communications* **8**, 16063 (2017).

Reviewers' Comments:

Reviewer #1 (Remarks to the Author):

Green et al. sufficiently addressed the main concerns of this reviewer. They examined the effect of substitution of AUG for CUG on RAN translation and provided the missing information regarding the effects of Sal003 and TM on eIF2alpha phosphorylation in HEK293 cells.

One remaining issue is that a repeat as short as 3xGA can initiate RAN translation in rabbit reticulocyte lysate (Fig. 1a). Although 70 repeat constructs have been used in most experiments presented in this paper, the results in Fig. 1a suggest that RAN translation is independent of repeat length and could potentially occur in normal condition. This goes against the pathophysiology of repeat-associated diseases where the length of repeats is a determining factor in disease severity. The authors should elaborate on this concern in the Discussion.

Reviewer #2 (Remarks to the Author):

The authors have responded well to comments and this reviewer finds that the current manuscript is acceptable. That said, the authors may want to consider several minor comments.

Minor comments

1. Figure 1 – are short repeats (i.e. from the 35 repeat of GAx in panel B) not toxic or are they just turned over more rapidly?
2. line 185 – Using bovine GAPDH as a common protein from which to gauge tRNA opthamality, it uses GCC for alanine about 50% of the time, CCG for proline 0% of the time and CGG for arginine 0% of the time. If this also represents the tRNA distribution in the tRNA added to the RRL, then one would anticipate seeing a much enhanced expression of the GA repeat relative to the other two.
3. line 366 – in both normal and disease state, what percent of the mRNAs are the repeat RNAs relative to the total mRNA population?
4. line 567 – Dr. Pelletier's name is misspelled.

Sunday, November 05, 2017

Dear Dr. Larochelle,

Please find attached our revised manuscript entitled "RAN translation at *C9orf72*-associated repeat expansions is selectively enhanced by the integrated stress response." We thank the reviewers for their careful reading of our manuscript, and have modified it in the following way to address their comments.

- 1) We added an additional paragraph to the discussion where we discuss RAN translation occurring at repeats length shorter than those known to be pathogenic.
- 2) We removed a sentence in our results section in which we suggest that differences in tRNA abundance likely do not contribute to the differences we observe in RAN translation levels across different reading frames.
- 3) We clarified that our expanded CGG and G₄C₂ repeat reporters are overexpressed in experiments where we assess stress granule formation and global translational suppression.
- 4) We corrected our misspelling of Dr. Jerry Pelletier's name.

With these modifications, we feel that our manuscript is now suitable for publication at *Nature Communications*.

Our specific responses to the reviewer comments are as follows (reviewer comments in italics).

Reviewer #1 (Remarks to the Author):

Green et al. sufficiently addressed the main concerns of this reviewer. They examined the effect of substitution of AUG for CUG on RAN translation and provided the missing information regarding the effects of Sal003 and TM on eIF2alpha phosphorylation in HEK293 cells.

One remaining issue is that a repeat as short as 3xGA can initiate RAN translation in rabbit reticulocyte lysate (Fig. 1a). Although 70 repeat constructs have been used in most experiments presented in this paper, the results in Fig. 1a suggest that RAN translation is independent of repeat length and could potentially occur in normal condition. This goes against the pathophysiology of repeat-associated diseases where the length of repeats is a determining factor in disease severity. The authors should elaborate on this concern in the Discussion.

We agree with the reviewer that our data in Fig. 1b, suggesting that 3 or 35 repeats can undergo RAN translation to generate short poly-GA products should be elaborated on, and we now include the following paragraph in our discussion.

"Interestingly, the mRNA reporters used in these studies generate a poly-GA product from only 3 or 35 G₄C₂ repeats, suggesting that the CUG codon, in good Kozak context, does not require an expanded repeat for use by the initiating ribosome. This is consistent with a report finding sparse neuronal DPR inclusions in a cognitively normal 84 year old woman harboring 30 *C9orf72* G₄C₂ repeats⁵⁶. The absence

of DPR accumulation in individuals with normal repeat sizes (<25) may indicate that these smaller species are rapidly cleared by cells, or that proper splicing and degradation of the intronic sequence containing the G₄C₂ repeat precludes its translation.”

Reviewer #2 (Remarks to the Author):

The authors have responded well to comments and this reviewer finds that the current manuscript is acceptable. That said, the authors may want to consider several minor comments.

Minor comments

1. Figure 1 – are short repeats (i.e. from the 35 repeat of GA_x in panel B) not toxic or are they just turned over more rapidly?

To our knowledge, there is no clear evidence of disease occurring as a result of G₄C₂ repeats shorter than 70. The lack of toxicity of shorter repeats is likely the result of a combination of factors. Studies that overexpress short DPRs in different model systems show that they can elicit toxicity, but typically to a lesser extent than larger DPRs¹. However, when expressed inefficiently at a low rate through RAN translation, it is possible that the cell is better able to clear small DPRs.

Consistent with our data showing RAN translation in the GA frame at 35 repeats, one report found a cognitively normal woman with 30 G₄C₂ repeats to have sparse neuronal DPR inclusions at the time of her death, at 84 years old². These DPR inclusions were less abundant than typically seen in large repeat carriers, suggesting that they were either produced less efficiently or aggregated at a slower rate².

Additionally, an important and unique feature about the G₄C₂ repeat is that it normally located within an intron. Therefore, although our data suggests that repeats of a normal length can produce a poly-GA product from linear, capped, and polyadenylated reporters, in individuals with normal repeat sizes, proper processing of the intron may prevent such species from being generated for ribosomes to act on.

We now include a paragraph in the discussion that addresses these different factors, in consideration of our data in Fig. 1b showing RAN translation at non-pathogenic repeat sizes.

2. line 185 – Using bovine GAPDH as a common protein from which to gauge tRNA opthamality, it uses GCC for alanine about 50% of the time, CCG for proline 0% of the time and CGG for arginine 0% of the time. If this also represents the tRNA distribution in the tRNA added to the RRL, then one would anticipate seeing a much enhanced expression of the GA repeat relative to the other two.

We agree with the reviewer that differences in tRNA abundance could contribute to differences in RAN translation levels across different reading frames. However, our data from Supplementary Fig. 1g, in which we use an AUG start codon to drive expression in the all three reading frames, indicates that our RRL system, supplemented with calf liver tRNA, has sufficient proline tRNA to generate the poly-GP product at levels equivalent to poly-GA.

Furthermore, like bovine GAPDH, human GAPDH also uses CCG and CGG for proline and arginine, respectively, 0% of the time. However, Pol III ChIP-seq data from human liver indicates that there is not a dramatic difference in the expression levels between the genes encoding the tRNAs that recognize the CCG and CGG codons for proline and arginine, respectively, and those encoding the tRNAs that recognize the alanine GGC codon³. Additionally, we observe similar frame-dependent differences in RAN

levels in RRL as we do across three other systems – two human cell lines and primary rodent neurons – suggesting that whatever is contributing to this difference is relevant in patients. However, as we have not directly assessed the levels of different tRNAs in our different systems, we have removed our statement regarding tRNA abundance previously in lines 182-184.

3. line 366 – in both normal and disease state, what percent of the mRNAs are the repeat RNAs relative to the total mRNA population?

In FXTAS, the CGG repeat is known to increase FMR1 transcript levels by up to 10-fold⁴. In C9ALS/FTD, while the expanded repeat is associated with reduced levels of C9orf72 transcripts, it is known to increase the number of transcripts containing the intron in which the repeat is located^{5, 6, 7}. Therefore, although we do not know what exact percent of cellular mRNAs the repeat-containing mRNAs make up, we know that it is significantly elevated in disease.

However, we acknowledge that in our study of the effects of CGG and G₄C₂ repeat expression on stress granule formation and global translational suppression, the expanded CGG and G₄C₂ repeat-containing constructs are being overexpressed in transfected cells, to levels likely higher than in disease. We have therefore modified our manuscript so that this overexpression is clear.

4. line 567 – Dr. Pelletier's name is misspelled.

We are grateful to the reviewer for catching this misspelling. It has now been corrected.

Sincerely,

Peter K. Todd, M.D., Ph.D.

Bucky and Patti Harris Professor

Associate Professor of Neurology

<https://sites.google.com/site/toddlabmichigan/>.

References

1. Mizielinska S, *et al.* C9orf72 repeat expansions cause neurodegeneration in Drosophila through arginine-rich proteins. *Science* **345**, 1192-1194 (2014).
2. Gami P, *et al.* A 30-unit hexanucleotide repeat expansion in C9orf72 induces pathological lesions with dipeptide-repeat proteins and RNA foci, but not TDP-43 inclusions and clinical disease. *Acta Neuropathol* **130**, 599-601 (2015).
3. Rudolph KL, *et al.* Codon-Driven Translational Efficiency Is Stable across Diverse Mammalian Cell States. *PLoS Genet* **12**, e1006024 (2016).
4. Tassone F, *et al.* Elevated FMR1 mRNA in premutation carriers is due to increased transcription. *RNA* **13**, 555-562 (2007).
5. Mori K, *et al.* The C9orf72 GGGGCC repeat is translated into aggregating dipeptide-repeat proteins in FTL/ALS. *Science* **339**, 1335-1338 (2013).
6. van Blitterswijk M, *et al.* Novel clinical associations with specific C9ORF72 transcripts in patients with repeat expansions in C9ORF72. *Acta Neuropathol* **130**, 863-876 (2015).
7. Niblock M, *et al.* Retention of hexanucleotide repeat-containing intron in C9orf72 mRNA: implications for the pathogenesis of ALS/FTD. *Acta Neuropathol Commun* **4**, 18 (2016).